# 21-nt phasiRNAs direct target mRNA cleavage in rice male germ cells

Pengfei Jiang[1,2,4], Bi Lian[1,2,4], Changzhen Liu[3], Zeyu Fu[1,2], Yi Shen[3], Zhukuan Cheng [3✉] & Yijun Qi [1,2✉]

In grasses, phased small interfering RNAs (phasiRNAs), 21- or 24-nucleotide (nt) in length, are predominantly expressed in anthers and play a role in regulating male fertility. However, their targets and mode of action on the targets remain unknown. Here we profile phasiRNA expression in premeiotic and meiotic spikelets as well as in purified male meiocytes at early prophase I, tetrads and microspores in rice. We show that 21-nt phasiRNAs are most abundant in meiocytes at early prophase I while 24-nt phasiRNAs are more abundant in tetrads and microspores. By performing highly sensitive degradome sequencing, we find that 21-nt phasiRNAs direct target mRNA cleavage in male germ cells, especially in meiocytes at early prophase I. These targets include 435 protein-coding genes and 71 transposons that show an enrichment for carbohydrate biosynthetic and metabolic pathways. Our study provides strong evidence that 21-nt phasiRNAs act in a target-cleavage mode and may facilitate the progression of meiosis by fine-tuning carbohydrate biosynthesis and metabolism in male germ cells.

[1] Center for Plant Biology, School of Life Sciences, Tsinghua University, Beijing 100084, China. [2] Tsinghua-Peking Center for Life Sciences, Beijing 100084, China. [3] State Key Laboratory of Plant Genomics, Institute of Genetics and Developmental Biology, Chinese Academy of Sciences, Beijing 100101, China. [4] These authors contributed equally: Pengfei Jiang, Bi Lian. ✉email: zkcheng@genetics.ac.cn; qiyijun@tsinghua.edu.cn

In plants, small RNAs (sRNAs) including microRNAs (miRNAs) and small interfering RNAs (siRNAs) are produced by Dicer-like (DCL) proteins and are associated with Argonaute (AGO) family proteins. They regulate target genes post-transcriptionally through cleaving mRNAs and repressing translation, or transcriptionally through directing DNA methylation[1–5].

Grasses contain a class of 21- or 24-nt phased siRNAs (phasiRNAs) abundantly expressed in anthers[6–10]. 21-nt phasiRNAs are most abundant in pre-meiotic anthers, whereas 24-nt phasiRNAs are most abundant in meiotic and post-meiotic anthers in maize and rice[6,10]. 21-nt phasiRNAs accumulate evenly in all anther cell layers, whereas 24-nt phasiRNAs are particularly enriched in tapetum and meiocytes in maize and Asparagus[6,11]. The generation of 21- and 24-nt phasiRNAs is usually triggered by miRNA2118 (miR2118)- and miR2775-directed cleavage of long non-coding precursor RNAs, respectively[12–14]. The cleavage products then serve as templates for RNA-dependent RNA polymerase 6 (RDR6) to produce double-stranded RNAs (dsRNAs)[15], which are processed into 21- and 24-nt phasiRNAs by DCL4 and DCL3b (also known as DCL5), respectively[9]. Many 21-nt phasiRNAs are associated with a germline-specific AGO protein MEL1 (MEIOSIS ARRESTED AT LEPTOTENE1)[12,16]. Knockout of DCL5 in maize and mutations of OsRDR6 and knockout of OsDCL4 or miR2118 in rice all cause male sterility[15,17–20]. Loss of MEL1 in rice causes arrest of male germ cell development at early meiotic stage[21]. Overaccumulation of 21-nt phasiRNAs derived from the photoperiod-sensitive genic male sterility 1 (Pms1) locus in rice causes photoperiod-sensitive male sterility[22]. These findings suggest that phasiRNAs might play critical roles in regulating germ cell meiosis. However, the targets of these reproductive phasiRNAs and the way they regulate their targets remain largely unknown.

Here, we use genomic approaches to demonstrate that 21-nt phasiRNAs are highly expressed in meiocytes at early prophase I and direct cleavage of hundreds of target genes. We propose that 21-nt phasiRNAs regulate male germ cell development through fine-tuning the expression of a large group of genes that together contribute to the progression of meiosis.

## Results

**Profiling of phasiRNAs in rice spikelets and purified male germ cells.** To profile phasiRNAs in male germ cells at different stages, we collected highly pure meiocytes at early prophase I (Fig. 1a, i), tetrads (Fig. 1a, ii), and microspores (Fig. 1a, iii and iv), respectively, from rice (Oryza sativa L. japonica cv. Nipponbare) by micromanipulation. Pre-meiotic spikelets (Fig. 1a, v), in which anthers are at stage 2 to stage 3 (Fig. 1a, vi)[23], meiotic spikelets (Fig. 1a, vii), in which germ cells are mostly at the early prophase of meiosis I, and embryos from germinated seeds, were also collected as controls. We performed low-input small RNA sequencing (sRNA-seq) using these samples (three biological replicates for each sample, Supplementary Data 1). sRNA-seq analysis revealed that in early prophase I meiocytes, the majority of the sRNAs were 21 nt in length (Supplementary Fig. 1a) and showed a 5′ nucleotide preference for cytosine (C) (Supplementary Fig. 1b). We next identified phasiRNAs produced in spikelets and germ cells at different stages of meiosis. A total of 11,810 21-nt phasiRNAs and 1689 24-nt phasiRNAs generated from 2173 and 195 loci, respectively, were identified (Fig. 1b, Supplementary Data 2, 3). 21-nt phasiRNAs generated from the Pms1 locus (annotated as 21PHAS1702 in this study) were detected (Supplementary Fig. 1c, Supplementary Data 2, 3). Among the phasiRNAs we identified, there were 899 21-nt phasiRNAs and 178 24-nt phasiRNAs derived from 317 and 58 loci, respectively, that have not been previously annotated (Supplementary Data 2, 3). As expected, the vast majority of 21-nt phasiRNA-producing loci

(21PHAS) were predicted to be targeted by miR2118 (Fig. 1b, Supplementary Data 2). More than half of 24-nt PHAS loci (24PHAS) were targeted by miR2275 (Fig. 1b, Supplementary Data 2). PhasiRNAs were barely detected in embryos and pre-meiotic spikelets (Fig. 1c, Supplementary Fig. 1d, Supplementary Data 3), but accounted for considerable portions of total sRNAs in early prophase I meiocytes, tetrads and microspores (Supplementary Fig. 1d). 21-nt phasiRNAs were much more abundant in early prophase I meiocytes than in tetrads and microspores and accounted for >70% of the total sRNAs in early prophase I meiocytes (Fig. 1d, Supplementary Data 3). The abundances of 24-nt phasiRNAs were higher in tetrads and microspores than in early prophase I meiocytes (Fig. 1c, Supplementary Data 3). Intriguingly, 21-nt phasiRNAs showed a very strong 5′ nucleotide preference for C in germ cells (Supplementary Fig. 1e).

**PhasiRNA accumulation in early prophase I meiocytes is largely dependent on OsRDR6 and MEL1.** In a screen for rice mutants that are sterile, we obtained the osrdr6-2 mutant (also called osrdr6-mei) that carries a point mutation in a conserved amino acid in the RDRP domain of RDR6[18,24] (Supplementary Fig. 2a–c) and the mel1-4 mutant that carries a deletion mutation in the first exon of MEL1, which introduces a premature stop codon (Supplementary Fig. 2d, e). We applied low-input sRNA sequencing to early prophase I meiocytes collected from osrdr6-2 and mel1-4 and their respective wild-type rice plants Zhongxian 3037 (3037) and Huanghuazhan (HHZ). Because osrdr6-2 is a point mutant, we first determined how much residual RDR6 activity is left in osrdr6-2 using the sRNA sequencing data. To this end, we examined the abundances of trans-acting siRNAs (tasiRNAs) produced from TAS3 loci in osrdr6-2. TasiRNAs are produced in somatic cells as well as reproductive cells from TAS3 loci and known to be OsRDR6-dependent[25–28]. There are four TAS3 loci in rice[29]. The abundance of tasiRNAs derived from TAS3a1 remained unaltered in osrdr6-2. TasiRNAs derived from TAS3a2 were hardly detected in wild-type 3037 and osrdr6-2, whereas those derived from TAS3b1 and TAS3b2 were reduced in abundance but not eliminated in osrdr6-2 (Supplementary Fig. 2f). These data suggest that there is a substantial amount of residual RDR6 activity in osrdr6-2 and osrdr6-2 is a weak allele. Next, differentially expressed phasiRNAs were identified between 3037 and osrdr6-2 and between HHZ and mel1-4 with a cutoff of greater than or equal to twofold change. We found that 73.4% and 24.4% of 21-nt phasiRNAs were OsRDR6-dependent and MEL1-dependent, respectively, with 22.2% being dependent on both OsRDR6 and MEL1, whereas 57.8% and 42.6% of 24-nt phasiRNAs were OsRDR6-dependent and MEL1-dependent, respectively, with 27.0% being dependent on both OsRDR6 and MEL1 (Fig. 2a, Supplementary Data 4). Overall, the abundances of 21-nt phasiRNAs were greatly reduced in osrdr6-2 and mel1-4 as compared with their respective wild-type rice plants (Fig. 2b, c, Supplementary Data 4). The abundances of 24-nt phasiRNAs were also reduced in the mutants, albeit to a lesser extent (Fig. 2b, d, Supplementary Data 4). Notably, in osrdr6-2, it is often the case that phasiRNAs within an entire PHAS locus had reduced expression (Fig. 2c, d, Supplementary Data 4). However, in mel1-4, only some phasiRNAs within a PHAS locus had reduced expression while others remained unaltered (Fig. 2c, d, Supplementary Data 4). Our results are consistent with the facts that OsRDR6 is required for general phasiRNA production[15], whereas MEL1 associates with and stabilizes some phasiRNAs[16].

**21-nt phasiRNAs direct target mRNA cleavage.** It is largely unknown what the targets and mode of action of reproductive phasiRNAs are. Most 21-nt phasiRNAs initiate with a 5′ C in

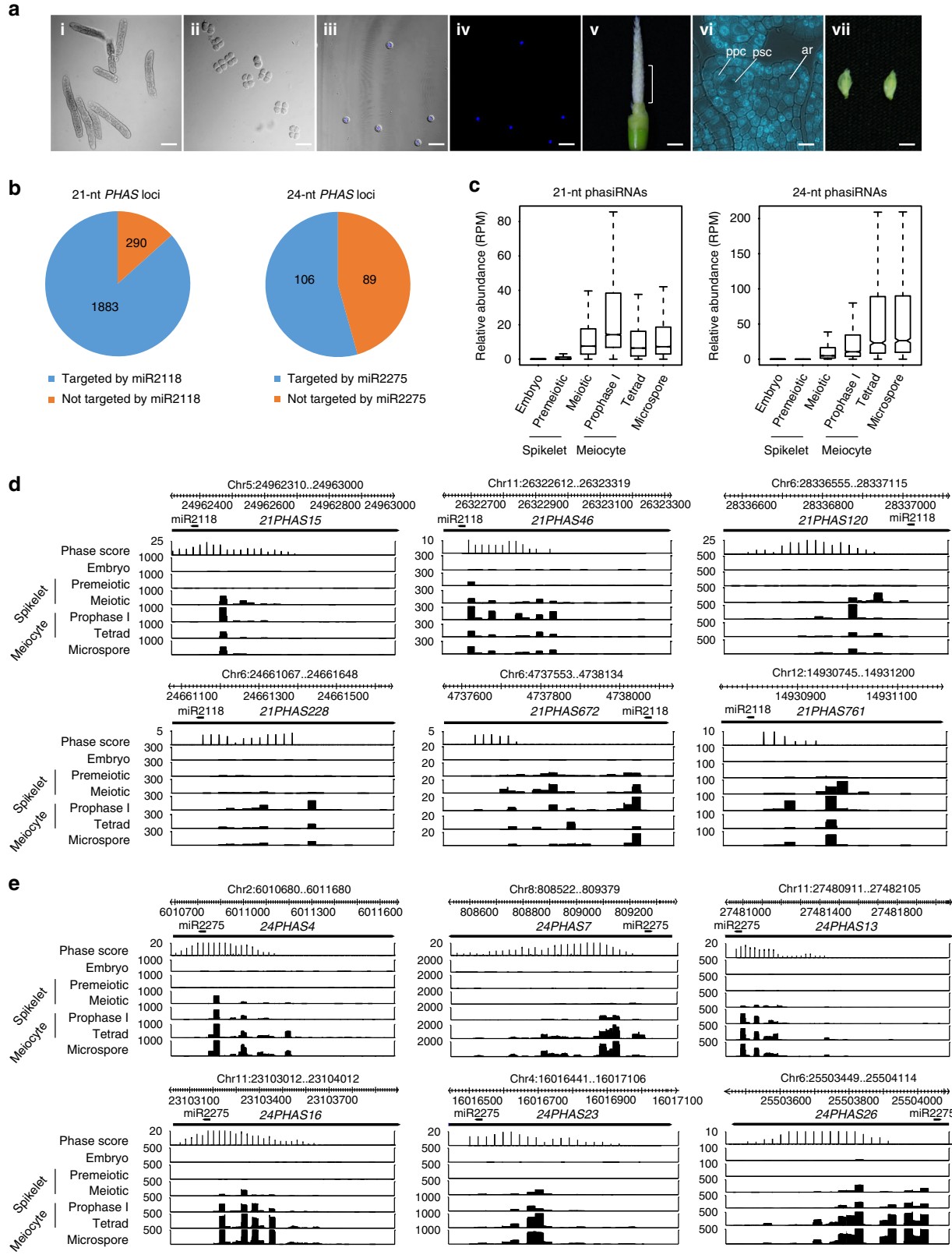

germ cells (Supplementary Fig. 1e). This predicts that they are very likely loaded into AGO5 subclade proteins, which are known to preferentially bind sRNA with a 5′ C[30]. As a matter of fact, many 21-nt phasiRNAs have been shown to be associated with MEL1 (OsAGO5c)[16]. In addition to MEL1, there are four AGO5s, namely OsAGO11/OsAGO5a, OsAGO14/OsAGO5b, OSAGO13/

OsAGO5d and OsAGO12/OsAGO5e (Supplementary Fig. 3a). Each of these OsAGO5s, except OsAGO5d, contains a conserved Asp-Asp-His (DDH) motif (Supplementary Fig. 3b), the catalytic triad required for cleavage activity[31], in their PIWI domain, raising the possibility that 21-nt phasiRNAs direct cleavage of target mRNAs like miRNAs.

**Fig. 1 Profiling of phasiRNAs in rice spikelets and purified male germ cells. a** Spikelets and male germ cells at different stages collected for phasiRNA profiling. i Worm-like clusters of meiocytes at early prophase I. ii Tetrads. iii Microspores, the bright-field and DAPI staining images are merged. iv DAPI-stained microspores. Scale bars in i–iv, 50 µm. v A panicle used for collecting pre-meiotic spikelets, only its lower part was taken. Scale bar, 0.2 cm. vi DAPI-stained cross-section of an anther from a pre-meiotic spikelet. *ppc* primary parietal cell, *psc* primary sporogenous cell, *ar* archesporial cell. Scale bar, 30 µm. vii Meiotic spikelets with meiocytes at early prophase I. Scale bar, 1 mm. **b** Pie chart showing the numbers of 21- and 24-nt *PHAS* loci targeted by miR2118 and miR2775, respectively. **c** Boxplot showing the relative abundances of 21- and 24-nt phasiRNAs in embryos, spikelets, meiocytes, and microspores. The central line of the box represents the median while two bounds represent 25% quartile and 75% quartile, respectively. The whisker represents 1.5× interquartile range of the lower or upper quartile. **d**, **e** Genome browser views of phasiRNA signals at the indicated 21- **d** and 24-nt **e** *PHAS* loci in embryos, spikelets, meiocytes, and microspores. Phase score was calculated within a nine-cycle window. The *y* axis represents sRNA abundance (RPM, Reads Per Million).

The Parallel Analysis of RNA Ends (PARE)/degradome sequencing method captures miRNA-induced cleavage products with a 5′ monophosphate and a poly (A) tail and is widely used for global identification of miRNA targets in plants[32,33]. Previous attempts to identify phasiRNA-directed cleavage events using PARE/degradome sequencing with bulk rice spikelets were unsuccessful[9,10]. We optimized the PARE/degradome sequencing protocol to allow use of low-input RNA (Supplementary Fig. 4). To compare the specificity and sensitivity of the low-input protocol with that of the regular PARE protocol[32], we applied the low-input protocol (with 5 ng and 50 ng of total RNA) and the PARE protocol (with 150 µg of total RNA) to 4-week-old rice seedlings and identified miRNA targets based on the specificity of AGO-mediated cleavage for a site between the 10th and 11th nucleotides from the 5′-end of the sRNA[34] (Supplementary Fig. 5a, b). In all, 131, 128, and 159 miRNA targets were identified by the low-input protocol with 5 ng and 50 ng of total RNA and the regular PARE protocol with 150 µg of total RNA, respectively. Among these, 127, 125, and 158 are previously validated miRNA targets in rice seedlings[35–37]. The miRNA targets identified by our low-input protocol and the regular PARE protocol overlapped to a great extent (Supplementary Fig. 5b). In all, 17 and 19 miRNA targets were identified by our low-input protocol with 5 ng and 50 ng of total RNA, but not identified by the regular PARE protocol. In all, 14 and 18 among the 17 and 19 miRNA targets are previously validated miRNA targets. These data suggest that our low-input protocol has a specificity and sensitivity comparable to those of the regular PARE method.

We then applied the low-input protocol to early prophase I meiocytes, tetrads, and microspores to detect miRNA- and phasiRNA-directed cleavage signals in these cells. Data analysis revealed that 74.8% of miR2118-targeted *PHAS* loci and 95.3% of miR2275-targeted *PHAS* loci were detected to have miR2118- and miR2275-directed cleavage signals, respectively, indicating a high quality of our degradome sequencing. We conducted a computational prediction of phasiRNA targets using criteria for miRNA target prediction and searched for cleavage products of the predicted targets in the degradome sequencing libraries. We identified 435 protein-coding genes and 71 transposable elements (TEs) as targets of 465 21-nt phasiRNAs derived from 413 loci in early prophase I meiocytes, tetrads and microspores (Fig. 3a, Supplementary Figs. 6–8, Supplementary Data 5). In total, 88.8%, 9.5%, and 1.7% of these *21PHAS* loci produced phasiRNAs targeting one, two and three or more genes, respectively (Supplementary Fig. 9, Supplementary Data 5). An unknown protein-coding gene *LOC_Os03g31410* was identified as the target of a phasiRNA (*21PHAS1702_287−*) derived from *Pms1* (Supplementary Fig. 6, Supplementary Data 5). The majority of 21-nt phasiRNA-guided cleavage products were detected only in early prophase I meiocytes (Fig. 3b, Supplementary Fig. 6, Supplementary Data 5). Using the same pipeline, we failed to identify any targets of 24-nt phasiRNAs, suggesting that directing target mRNA cleavage is unlikely a mode of action common to both 21-nt and 24-nt phasiRNAs.

To further confirm our degradome sequencing results and determine the dependence of 21-nt phasiRNA-directed cleavage on OsRDR6 and MEL1, we performed degradome sequencing of early prophase I meiocytes of *osrdr6-2* and *mel1-4* mutants and their corresponding wild-type lines 3037 and HHZ. 21-nt phasiRNA-directed cleavage products from 131 (121 protein-coding genes and 10 TEs) and 133 targets (121 protein-coding genes and 12 TEs) were detected in 3037 and HHZ, respectively (Supplementary Data 6). The abundances of degradome tags at phasiRNA-guided cleavage sites of most targets were markedly reduced in *osrdr6-2* and *mel1-4* (Supplementary Fig. 10a, b, Supplementary Data 6), suggesting that most cleavage events are dependent on OsRDR6 and MEL1.

To examine whether phasiRNAs mediating target cleavage are more stable than others, we compared the abundances of phasiRNAs with targets with those without targets in prophase I meiocytes of Nipponbare. The results showed that phasiRNAs with targets were more abundant than others (Supplementary Fig. 11a). PhasiRNAs with targets were decreased to a greater extent than the ones without targets in abundance in the *mel1-4* mutant (Supplementary Fig. 11b). We also performed analyses using published sRNA-seq data from immunopurified MEL1[16]. We found that 71% of phasiRNAs with targets were MEL1-associated, while a lower percentage (50%) of phasiRNAs without targets were MEL1-associated (Supplementary Fig. 11c). Furthermore, phasiR-NAs with targets were more abundant in immunopurified MEL1 (Supplementary Fig. 11d). These findings suggest that phasiRNAs mediating target cleavage are more stable, presumably because of their association with MEL1.

DMC1 is one of the recombinases in eukaryotic cells homologous to *Escherichia coli* RecA[38], with the other being Rad51. *OsDMC1A* (*LOC_Os12g04980*), which acts redundantly with *OsDMC1B* to regulate synapsis in meiosis[39], was detected to be a target of 21-nt phasiRNA (*21PHAS1857_165+*) (Supplementary Fig. 7, Supplementary Data 5), suggesting that 21-nt phasiRNAs may be directly involved in the regulation of meiotic gene expression. To understand the general role of 21-nt phasiRNAs in germ cell meiosis, we performed Gene Ontology (GO) analysis on 253 protein-coding genes targeted by 21-nt phasiRNAs in early prophase I meiocytes of Nipponbare (Fig. 3c). The result showed that these genes were enriched for carbohydrate biosynthetic and metabolic processes, suggesting that 21-nt phasiRNAs play an important role in regulating carbohydrate biosynthesis and metabolism in rice male meiocytes.

**21-nt phasiRNAs regulate target gene expression in early prophase I meiocytes.** Next, to study the effects of phasiRNAs on gene expression, we performed low-input mRNA sequencing (mRNA-seq) in early prophase I meiocytes collected from *osrdr6-2* and *mel1-4* and their respective wild-type rice plants. Among 253 protein-coding genes targeted by 21-nt phasiRNAs in early prophase I meiocytes of Nipponbare, 33 genes were not detected by mRNA-seq. Among the remaining 220 genes, 23 genes

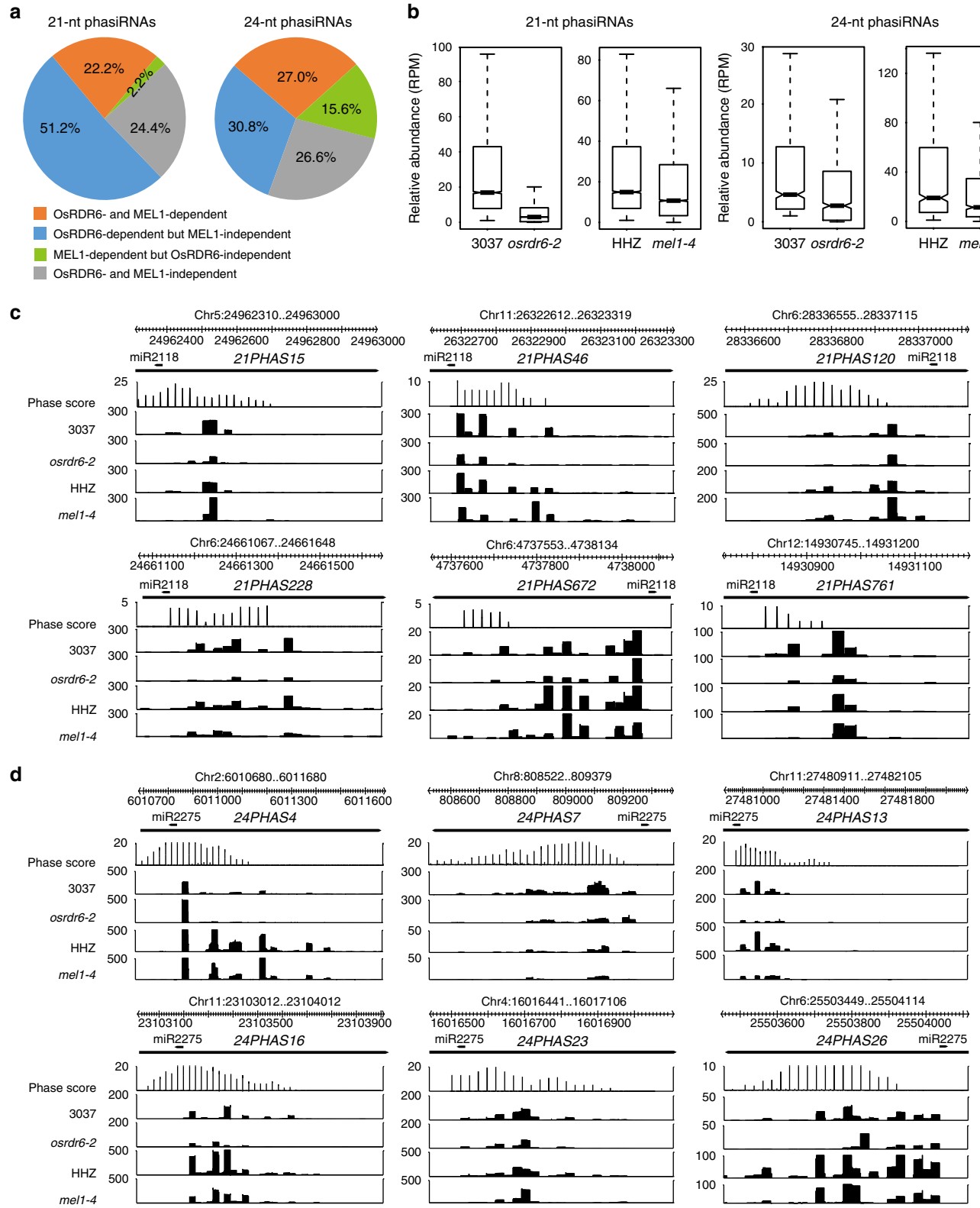

**Fig. 2 PhasiRNA accumulation in early prophase I meiocytes is largely dependent on OsRDR6 and MEL1. a** Pie charts showing the percentages of phasiRNAs whose accumulation is dependent or independent on OsRDR6 or MEL1. **b** Boxplots showing the relative abundances of 21- and 24-nt phasiRNAs in prophase I meiocytes of the *osrdr6-2* and *mel1-4* mutants and their respective wild-type plants. The central line of the box represents the median while two bounds represent 25% quartile and 75% quartile, respectively. The whisker represents 1.5× interquartile range of the lower or upper quartile. **c, d** Genome browser views of phasiRNA signals at the indicated 21- **c** and 24-nt **d** *PHAS* loci in early prophase I meiocytes of the *osrdr6-2* and *mel1-4* mutants and their respective wild-type plants.

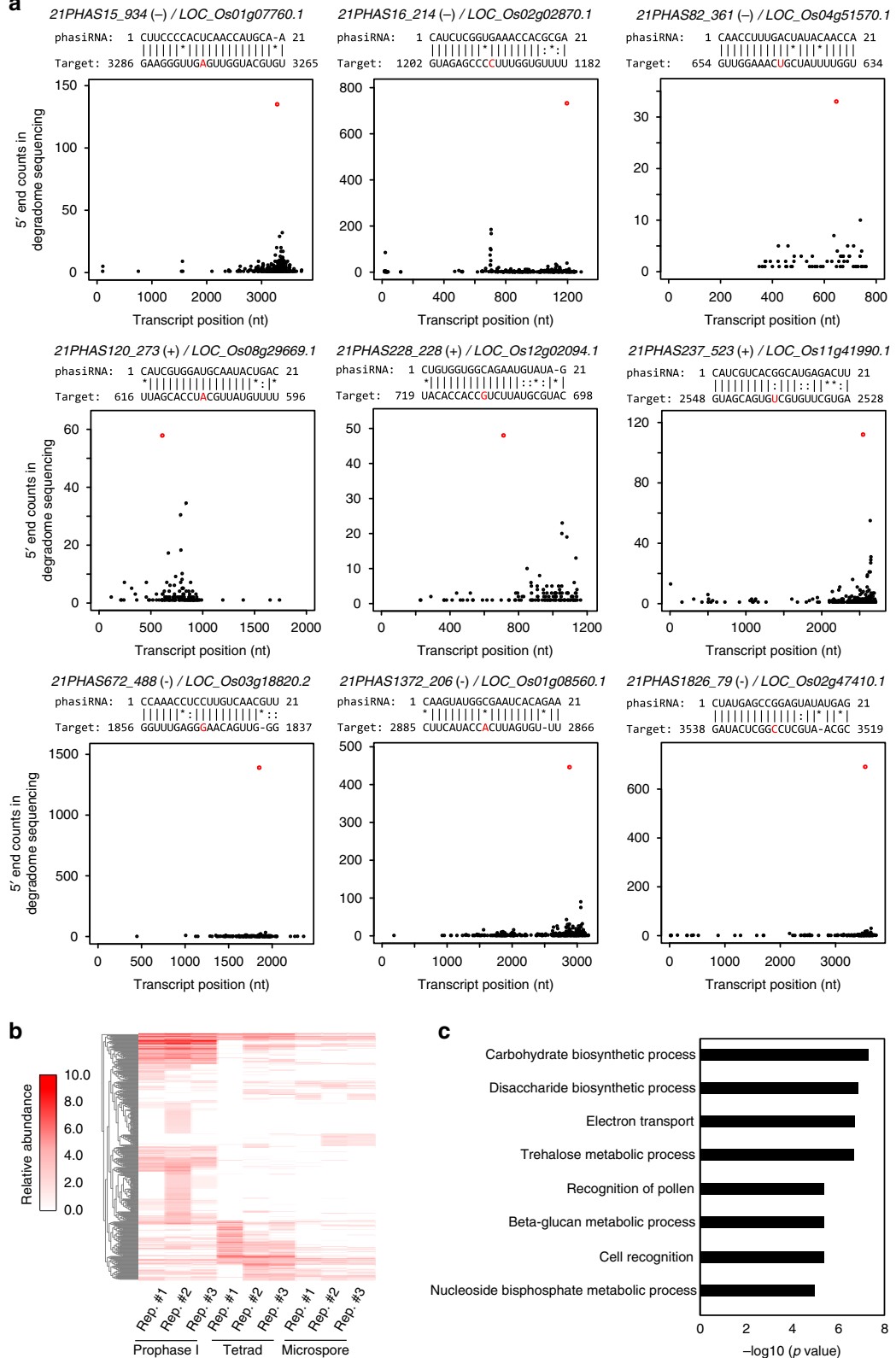

**Fig. 3 21-nt phasiRNAs direct target mRNA cleavage. a** Plots showing the distribution of the degradome tags along representative 21-nt phasiRNA targets. The x axis represents the nucleotide position of each target mRNA (5′–3′). The y axis represents the relative frequency of degradome tags. The red dot represents degradome tag at +1 position of 21-nt phasiRNA-guided cleavage site. **b** Heatmap showing the abundances of degradome tags at +1 positions of 21-nt phasiRNA-guided cleavage sites in early prophase I meiocytes, tetrads and microspores. Colors represent RP10M (Reads Per 10 Million) values that are log2-transformed after adding a pseudocount of 1. **c** Gene ontology enrichment analysis of the validated 21-nt phasiRNA targets in early prophase I meiocytes. GO biological process terms significantly enriched in the target genes are shown. The P values were calculated by using Fisher's exact test.

targeted by 24 21-nt phasiRNAs and 22 genes targeted by 22 21-nt phasiRNAs were downregulated in *osrdr6-2* and *mel1-4*, respectively, with a cutoff of ≥1.5-fold change and *q* value ≤ 0.05. 51 genes targeted by 54 21-nt phasiRNAs and 25 genes targeted by 30 21-nt phasiRNAs were upregulated in *osrdr6-2* and *mel1-4*, respectively, with a cutoff of ≥1.5-fold change and *q* value ≤ 0.05. Among them, 11 genes targeted by 13 21-nt phasiRNAs were upregulated in both *osrdr6-2* and *mel1-4* (Fig. 4a-c, Supplementary Data 7). In addition, there are 9 and 23 genes, which showed significant increase but did not have fold changes above 1.5 in *osrdr6-2* and *mel1-4*, respectively (Supplementary Data 7). The percentage of upregulated targets in total phasiRNA targets was significantly higher than that of upregulated genes in all expressed genes. Thus, phasiRNA targets appeared to be enriched among upregulated genes (Supplementary Fig. 12a, b). GO analysis revealed that the derepressed 21-nt phasiRNA targets also showed enrichment for the carbohydrate biosynthetic pathway (Supplementary Fig. 13), further supporting that 21-nt phasiRNAs regulate carbohydrate-related biological processes. Together, our results suggest that 21-nt phasiRNAs regulate gene expression in rice male meiocytes.

## Discussion

In this study, we find that 21-nt phasiRNAs are extremely abundant in meiocytes at early prophase I, whereas 24-nt phasiRNAs are more abundant in tetrads and microspores (Fig. 1). This shows consistency with the previously defined temporal accumulation pattern of phasiRNAs in anthers[6,10]. 21-nt phasiRNAs in male germ cells show very strong 5′ nucleotide preference for C, coincident with the specific expression of *MEL1* in germ cells[21]. By performing low-input degradome sequencing in purified germ cells, we obtained strong evidence that hundreds of 21-nt phasiRNAs direct cleavage of hundreds of target mRNAs (Fig. 3), like miRNAs. The repertoire of high-confidence phasiRNA targets generated in this study will provide a rich resource for the community to further investigate the biology of phasiRNAs in plants.

It was noticed that in *osrdr6-2* and *mel1-4* mutants, many targets did not show derepressed expression (≥1.5-fold change). There are several reasons that may be the causes of this. First, the *osrdr6-2* mutant is a weak allele. The expression of some targets may not be affected by the *osrdr6-2* mutation. Second, in gene regulatory networks, there are many feedback loops. The genes that are post-transcriptionally derepressed could be transcriptionally repressed to maintain gene expression homeostasis. Third, phasiRNAs may regulate target gene expression using other modes of action including translation repression. Fourth, the importance of these phasiRNAs could lie in their capability to target many genes. They do not necessarily induce high fold-changes of target gene expression.

An miRNA usually can only regulate a single gene or a gene family with several members. Through triggering the production of 21-nt phasiRNAs from over 2000 loci, miR2118 greatly extends its regulatory range. The most enriched GO terms of the target genes are associated with carbohydrate biosynthesis and metabolism (Fig. 3). Interference with carbohydrate production leads to male sterility[40,41], suggesting that carbohydrate biosynthesis and metabolism are critical for male germ cell development. We thus propose that 21-nt phasiRNAs act as a group to regulate biosynthesis and metabolism of carbohydrates, which serve as nutrients and signaling molecules, to prepare the cells for the progression through meiosis. Interestingly, 21-nt phasiRNAs were also validated to direct cleavage of dozens of TE transcripts (Supplementary Data 5). The biological role of these 21-nt phasiRNA-guided cleavage events remains to be studied. Although we demonstrate that 21-nt phasiRNAs direct target cleavage, we

did not find evidence for target mRNA cleavage directed by 24-nt phasiRNAs. It is important to explore whether and how they function to contribute to male germ cell development in the future.

piRNAs are a class of small non-coding RNA specifically expressed in animal cells. phasiRNAs and piRNAs evolved independently, they use different pathways for biogenesis and associate with different effector proteins. However, phasiRNAs are analogous to piRNAs in their enrichment in reproductive organs and involvement in fertility regulation[13,14,42]. piRNAs mediate cleavage of meiotic transcripts derived from protein-coding genes and TEs to regulate spermatogenesis[43]. Here, we demonstrate that phasiRNAs have the same mode of action. Our findings underscore evolutionary conservation of sRNA actions.

## Methods

**Plant materials and growth conditions.** Rice seeds of Nipponbare (*japonica*), Zhongxian 3037 (*indica*), Huanghuazhan (*indica*) as well as *osrdr6-2* and *mel1-4* were sterilized with 5% NaClO containing 0.05% Tween-20 for 45 min. The seeds were then germinated and grown in either a paddy field in Changping District, Beijing, China (116.42°, 40.10°) from May to October or in a greenhouse with 40% humidity and daily cycles of 14 h of light at 32 °C and 10 h of dark at 26 °C. Four-week-old seedlings of Nipponbare (*japonica*) were harvested for degradome sequencing using low-input and regular PARE protocols. Tillers of different ecotypes and genotypes at a proper stage were harvested for collection of spikelets and male germ cells. The collected spikelets and male germ cells were used for low-input sRNA sequencing, degradome sequencing and mRNA sequencing.

**Collection of embryos, spikelets, and male germ cells.** For examining the dynamic expression of phasiRNAs in rice male germ cells, Nipponbare rice plants were used for collection of embryos, spikelets and male germ cells. Seeds without glumes were sterilized and incubated at 28 °C for 24 h for dissecting and collecting embryos. For collection of pre-meiotic spikelets, panicles that were ~1 cm long were chosen and lower half parts of the panicles were collected. For collection of meiotic spikelets with germ cells at early prophase of meiosis I, tillers whose flag leaf collar was about 8 cm under the collar of the penultimate leaf were chosen and 1.5–2 mm long spikelets were collected. Meiocytes at early prophase I were collected as previously described[44] with minor modifications. In detail, meiotic spikelets collected above were put on a slide. The anthers were quickly dissected out and collected using dissecting needles. Then one drop of 1× phosphate-buffered saline (PBS) was added. Because germ cells at this stage produce callose, they stick together and form worm-like cell clusters. Gentle pressure was applied to the anthers to allow the separation of worm-like cell clusters from anther walls. After the addition of one drop of 1× PBS, these cells were collected with capillary glass pipettes without taking any somatic cell debris. For rice in Zhongxian 3037 (3037) and Huanghuazhan (HHZ) backgrounds, meiocytes at early prophase I were collected from 3–4 mm long spikelets, which were from tillers with ~5-cm distance between flag leaf collar and penultimate leaf collar. For collection of tetrads and microspores, tillers whose flag leaf collar and penultimate leaf collar aligned were chosen and branches on which spikelets were 4–6 mm long were taken. The top spikelet was discarded and the remaining spikelets were put on a slide according to their developmental orders on a branch[45]. Anthers were dissected out and gentle pressure was applied to allow the release of germ cells. Cells with tetrad shapes were collected as tetrads. Cells released from the next spikelet were usually microspores and were collected after assessment by DAPI staining. The collected germ cells were washed twice with 1× PBS, resuspended in 100 μL of TRIzol and then stored in 0.2 mL LoBind tubes (Axygen, PCR-02-L-C) at −80 °C for future RNA extraction.

**RNA extraction.** Total RNA was extracted using TRIzol (Takara, 9109). MgCl₂ (final concentration: 0.2 M) was added to prevent loss of sRNAs with low GC content[46]. The extracted RNA was stored at −80 °C in LoBind tubes (Eppendorf, 022431021).

**Small RNA sequencing.** Libraries for sRNA-seq were constructed as previously described[47] with minor modifications. In brief, 3′ adaptor, DNA oligonucleotide with 5′ adenylation, was obtained using 5′ DNA Adenylation Kit (NEB, E2601L). Total RNA was ligated to 3′ adaptor using T4 RNA ligase 2, truncated KQ (NEB, M0373L). Following annealing with reverse transcription primer, RNA was ligated to 5′ RNA adaptor using T4 RNA ligase 1 (NEB, M0204L). After reverse transcription and PCR amplification, 135–160-bp products were gel-purified. Libraries were pair-end sequenced on an Illumina HiSeqX-Ten platform by Annoroad Gene Technology (Beijing).

**Small RNA sequencing data analysis.** Low-quality reads were removed and 3′ adapter sequences were trimmed using cutadapt2.7. Clean sRNA reads in sizes of 18–30 nt were mapped to the rice reference genome (MSU v7) using bowtie,

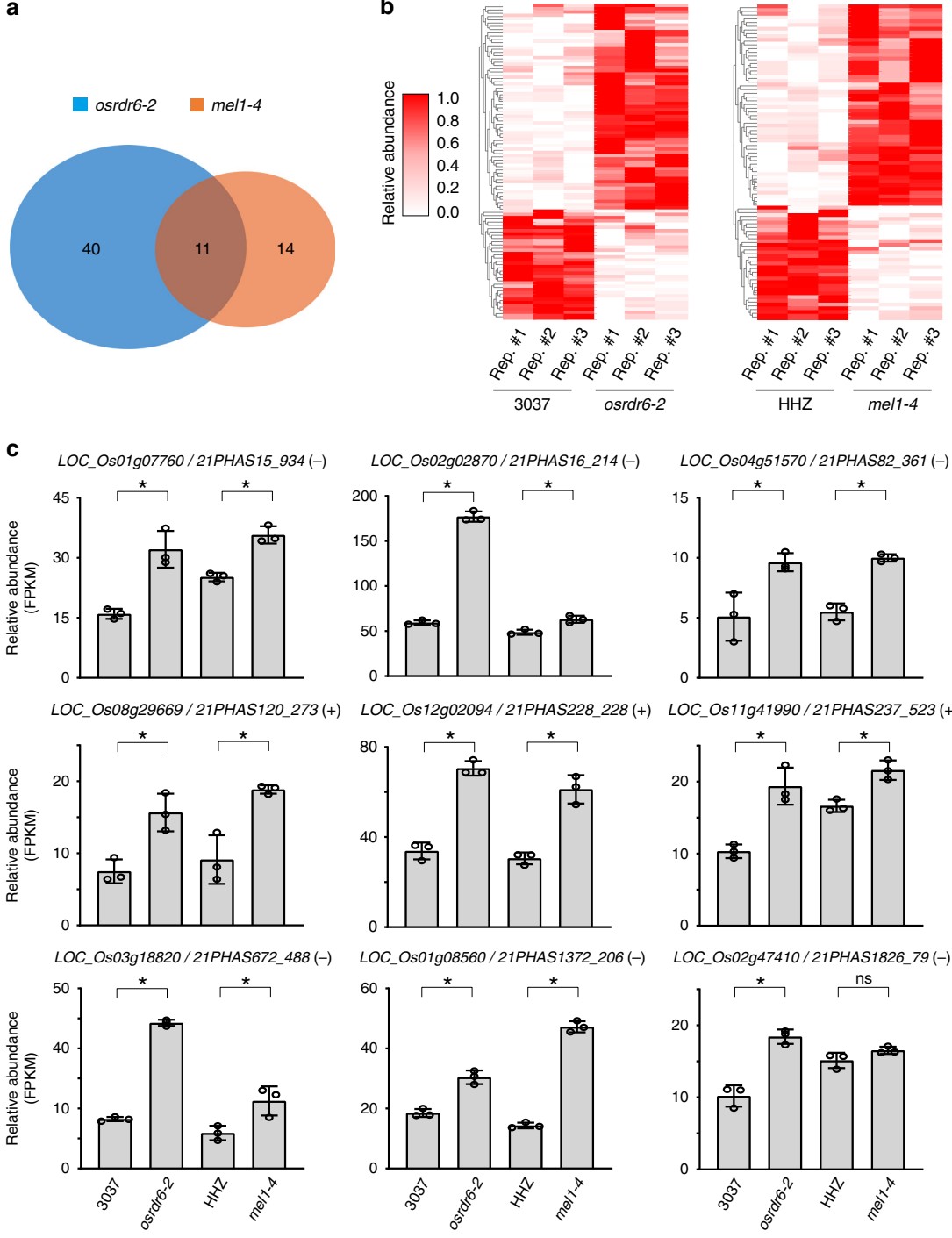

**Fig. 4 21-nt phasiRNAs regulate target gene expression in early prophase I meiocytes. a** Venn diagram showing the overlap of upregulated 21-nt phasiRNA targets in early prophase I meiocytes of *osrdr6-2* and *mel1-4*. **b** Heatmap showing the relative expression levels of upregulated and downregulated targets in early prophase I meiocytes of *osrdr6-2* and *mel1-4* and their respective wild-type plants. Colors represent the fractional density across the row of log2-transformed (FPKM+1) values. **c** Relative expression levels of representative upregulated targets as determined by mRNA-seq. The error bars represent standard deviation of three biological replicates. Asterisks indicate significant differences between mutant and the corresponding wild-type plants. *q value ≤0.05. q values were calculated by Cuffdiff.

allowing no mismatches. 21-nt and 24-nt *PHAS* loci were identified and characterized by PhaseTank and Unitas as previously described[48,49] using all 21-nt and 24-nt genome-matched reads from spikelets and male germ cells. *PHAS* loci identified by both of these two programs were considered as real *PHAS* loci. For examining the dependence of phasiRNA accumulation on MEL1 or OsRDR6, the read counts of each phasiRNA from three biological replicates of wild-type and mutant plants were compared using edgeR[50]. The resulting *P* value was adjusted to estimate false discovery rate (FDR). PhasiRNAs with a ≥2-fold reduction in the mutant and an FDR ≤0.05 were considered to be dependent on OsRDR6 or MEL1.

**Degradome sequencing.** The degradome libraries (the regular PARE protocol) were constructed as previously described[32]. In brief, polyadenylated RNAs were purified from total RNA and those containing a 5′ monophosphate were ligated to an RNA adapter with a MmeI recognition site at its 3′-end. After ligation, first-strand cDNA was reversely transcribed using oligo-d(T) primer and then amplified by 7 cycles of PCR. The PCR products were purified and digested with MmeI. The products of the digestive process, a fragment containing the 5′ adapter sequence and 20 bp of gene-specific sequence ligated to the 5′ adapter, were selected based on their sizes and gel-purified. The purified products were ligated to 3′ double-

stranded-DNA adapter. The ligation products were gel-purified and amplified by 20 cycles of PCR. The PCR products were gel-purified and sequenced on Illumina HiSeqX-Ten platform by Annoroad Gene Technology (Beijing).

For degradome sequencing using low-input RNA, a low-input protocol was developed based on the regular PARE protocol (Supplementary Fig. 4). In detail, 5′ RNA adaptor (5′-GUUCAGAGUUCUACAGUCCGAC-3′) was first heated at 75 °C for 3 min and then immediately chilled on ice for 2 min. For 5′ RNA adapter ligation, total RNA (3 μL) was added to a solution composed of 1 μL of T4 RNA ligase buffer, 1 μL of 10 mM ATP, 0.5 μL of Ribonuclease inhibitor (Promega, N2511), 0.5 μL of 10 μM treated RNA adaptor, 3 μL of PEG8000 (50% v/v), and 1 μL of T4 RNA ligase 1 and incubated at room temperature for 1 h. Then 1 μL of 10 mM dNTP and 0.5 μL of 10 μM reverse transcription primer Oligo(dT)VN (5′-CGAGCACAGAATTAAT ACGACT$_{(30)}$VN-3′) were added. After incubating at 75 °C for 5 min and chilling on ice for 2 min, 5 μL of first-strand RT buffer, 5 μL of 5 M betaine (Sigma, 61962), 1.25 μL of 0.1 M DTT, 0.9 μL of 1/6 M MgCl$_2$, 0.5 μL of Ribonuclease inhibitor, and 1 μL of SuperScript III reverse transcriptase (Invitrogen, 18080-085) were added for reverse transcription. The condition for reverse transcription was 44 °C for 1 h followed by 15 cycle of 50 °C for 2 min and 44 °C for 2 min. After reverse transcription, the products were purified with 2× AMPure XP beads (v/v). Then the products (20 μL) were pre-amplified using GXL DNA polymerase (Takara, R050A) with 5′ biotin-labeled forward primer F1 (5′-bio-GTTCAGAGTTCTACAGTCCGAC-3′) and reverse primer R1 (5′-CGAGCACAGAATTAATACGACT-3′) in a solution composed of 10 μL of 5× GXL buffer, 4 μL of 2.5 mM dNTP, 1.5 μL of 10 μM F-primer, 1.5 μL of 10 μM R-primer, 2 μL of GXL DNA polymerase, 11 μL of ddH$_2$O. The PCR was performed at 98 °C for 2 min followed by 20 cycle of 98 °C for 10 s, 60 °C for 15 s, 68 °C for 3 min and then 68 °C for 5 min. After PCR, 1 μL of Exonuclease I (NEB, M0293L) and 5.7 μL of 10× Exonuclease I buffer were added to the sample and the mixture was incubated at 37 °C for 1 h. Then 19 μL of 4× binding buffer (40 mM Tris-HCl, pH 8.0, 2 mM EDTA, 4 M NaCl) was added and the mixture was incubated with Dynabeads MyOne Streptavidin C1 beads (Invitrogen, 65001) at room temperature for 30 min with rotation. The beads were washed once with 1× TWB buffer (10 mM Tris-HCl, pH 8.0, 0.5 M EDTA, 1 M NaCl, 0.05% Tween-20 v/v) and 3 times with 1× EBT buffer (10 mM Tris-HCl, pH 8.0, 0.02% Triton X-100 v/v) and then digested with MmeI (NEB, R0637L) in a solution composed of 3 μL of 10× NEB buffer 2, 3 μL of 0.5 mM SAM, 1 μL of rSAP (NEB, M0371L), 2 μL of Mme I and 21 μL of ddH$_2$O at 37 °C for 2 h with rotation. The beads were washed once with 1× TWB buffer and 3 times with 1× EBT buffer. The DNA products bound to the beads were ligated to 3′ DNA adaptor in a solution composed of 30 μL of 2× quick ligase reaction buffer, 1 μL of 10 μM adaptor, 1 μL of quick ligase (NEB, M2200L) and 25 μL of ddH$_2$O at room temperature for 30 min with rotation. The 3′ DNA adapter was prepared by annealing of two DNA oligos (5′-pATGGAATTCTCGGGTGCCAAGGC-3′ and 5′-GCCTTG GCACCCGAGAATTCCATNN-3′) in 1× annealing buffer (10 mM Tris-HCl pH 7.5, 0.1 mM EDTA, 50 mM NaCl). The beads were washed once with 1× TWB buffer and three times with 1× EBT buffer and resuspended in 30 μL of 0.05% Triton X-100. The samples were incubated at 80 °C for 30 min with rotation to elute the DNA products. Then the DNA products were subjected to final PCR amplification using GXL DNA polymerase and primers F2 (5′-AATGATACGGCGACCACCGAGATCTACACGT TCAGAGTTCTACAGTCCGA-3′) and R2 (5′-CAAGCAGAAGACGGCATACGA GAT-Index-GTGACTGGAGTTCCTTGGCACCCGAGAATTCCA-3′). PCR was performed at 98 °C for 2 min followed by 22 cycle of 98 °C for 10 s, 60 °C for 15 s, 68 °C for 20 s and then 68 °C for 5 min. The final products were resolved on a polyacrylamide gel electrophoresis gel and ~140-bp bands were excised and purified for sequencing on an Illumina HiSeqX-Ten platform by Annoroad Gene Technology (Beijing).

**Degradome sequencing data analysis.** All of annotated miRNAs/phasiRNAs were aligned to rice cDNA sequences downloaded from rice genome database (MSU v7) using psRobot_tar (-ts 3.5 –fp 2 –tp 17)[51]. Mispair score was calculated by psRobot using the following rules: (1) mismatches, gaps or bulges incur a penalty of +1; (2) G:U pairs incur a penalty of +0.5; (3) penalty scores outside the core base-paired region (2–17) are reduced by half. Genes (including TEs) that pair with miRNAs/phasiRNAs with mispair scores ≤3.5 were selected as an initial pool of potential miRNA/phasiRNA targets. The 5′-ends of degradome tags derived from miRNA/phasiRNA cleavage fragments should precisely match the 10th nucleotide of miRNA/phasiRNA complementary sites. Such tags were searched in the degradome libraries using psRobot_deg[51]. All hit transcripts are categorized based on the abundance of the diagnostic cleavage tag relative to the overall profile of degradome tags matching the target essentially as previously described[33,52]. Category 4, 1 degradome tag at the cleavage site; Category 3, degradome tags at the cleavage site >1, but ≤ the average number of degradome tags on the transcript; Category 2, degradome tags at the cleavage site >1, above the average but not the maximum on the transcript; Category 1, degradome tags at the cleavage site >1, equal to the maximum on the transcript, but there are >1 sites at maximum value; Category 0, >1 degradome tags, equal to the maximum on the transcript, and there is only 1 site at the maximum value. Category 0 or Category 1 transcripts that have ≥4 degradome tags at the cleavage site, which account for ≥10% of the total degradome tags along the transcript, were annotated as miRNA/phasiRNA targets. Gene ontology enrichment of phasiRNA targets in early prophase I meiocytes was performed by plantGSEA[53] using all expressed genes in early prophase I meiocytes as a background control.

**mRNA sequencing.** Libraries for mRNA-seq were constructed using the Smart2-seq method[54]. In brief, total RNA was used for reverse transcription and cDNA second strand synthesis using SuperScript II reverse transcriptase (Invitrogen,18064-14) and template-switching oligos. After PCR preamplification and purification with AMPure XP beads (Beckman, A63880), the products were subjected to fragmentation and adaptor ligation using TruePrepTM DNA Library Prep Kit (Vazyme, TD503). PCR amplification was then performed with index-containing primers and 300–800-bp products were purified with AMPure XP beads. Libraries were pair-end sequenced on an Illumina HiSeqX-Ten platform by Annoroad Gene Technology (Beijing).

**mRNA sequencing data analysis.** Low-quality reads were removed and adapters were trimmed using Trimmomatic (v0.36)[55]. Clean RNA reads were then mapped to the reference genome (MSU v7) with default parameters using TopHat v2.1.1. The expression level of each gene was normalized to fragments per kilobase of transcript per million mapped reads. Differential gene expression analysis was performed using Cuffdiff[56]. Genes with a ≥1.5-fold change and an FDR ≤0.05 were identified as differentially expressed genes.

**Reporting summary.** Further information on research design is available in the Nature Research Reporting Summary linked to this article.

## Data availability
Small RNA, degradome, and mRNA sequencing data sets generated in this study can be found in the NCBI Gene Expression Omnibus under accession number GSE149800. A reporting summary for this article is available as a Supplementary Information file. All other data that support the findings of this study are available from the corresponding author upon request. Source data are provided with this paper.

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

## Acknowledgements

We are grateful to Jiayang Li and Kun Yuan (Chinese Academy of Sciences) for kind help with rice cultivation and to W. Xie, Z. Xiong, and Y. Zhang (Tsinghua University) for sharing library construction protocols. This work was supported by grants from National Science Foundation of China (grant no. 31788103 and 31421001) and National Key R&D Program of China (Grant No. 2016YFA0500800) to Y.Q. Y.Q. is a visiting investigator of the CAS Center for Excellence in Molecular Plant Sciences.

## Author contributions

P.J. and Y.Q. conceived the project and designed the experiments, P.J. conducted the experiments with assistance from Z.F., C.L. provided plant materials, B.L. performed bioinformatic analysis. P.J., B.L., Z.C., and Y.Q. analyzed the data, Y.Q. wrote the manuscript. All authors discussed the results and made comments on the manuscript.

## Competing interests

The authors declare no competing interests.
