## [Peer Review File · Nature Communications]

Reviewers' comments:

Reviewer #1 (Remarks to the Author):

Jiang et al. make an important step forward to understanding the biological functions of reproductive phasiRNAs. By using isolated meiocytes from rice and integrating sRNA-seq with mRNAseq and degradome sequencing adopted for low input material, they identified cleavage of specific targets in early prophase I, mediated by 21nt phasiRNAs, namely 554 protein coding genes and 102 transposons. This is of note since most previous studies in maize and rice did not find target genes.

An additional interesting finding is that no target gene cleavage was found for 24nt phasiRNAs which might point to diverse functions of those otherwise similar classes of reproductive phasiRNAs. The manuscript is well written and precise, figures support the drawn conclusions, and the data is sufficient for the claims made. Addressing my questions and comments below will help to get to a final version that is clear and connected with relevant other knowledge.

The study is of high interest to the field of plant reproductive phasiRNA research. However, the manuscript would benefit from connecting the findings with previous studies to put them into contexts:

- Tamim et al 2018. Cis-directed cleavage and nonstoichiometric abundances of 21-nucleotide reproductive phased small interfering RNAs in grasses: This paper also found that specific phasiRNAs are more stable than others. Are those related with the ones mediating cleavage? Same question for the results from the MEL1 paper cited and the current mel1-data – are the MEL1-stabilized phasiRNAs overlapping the ones mediating cleavage?
- Fan et al 2016. PMS1T, producing phased small-interfering RNAs, regulates photoperiod-sensitive male sterility in rice; and Wang et al 2011. Comparative transcriptomes profiling of photoperiod-sensitive male sterile rice nongken 58S during the male sterility transition between short-day and long-day: These papers describe an important rice 21phasi locus that confers conditional male sterility, and identify target genes or transcriptional changes. Does the present dataset include this finding and/or can confirm or extend it?
- Put into wider context: very similar to animal piRNAs (silencing of meiotic genes and TEs): Goh et al 2015. piRNA-directed cleavage of meiotic transcripts regulates spermatogenesis

I am not completely clear about the following points which might be addressed:

- Numbers that seem not to match to me:
 - o Only about half of the 24nt phasi loci are targeted by miR2775 but 94.5% had miR2275-directed cleavage patterns?
 - o Concerning protein coding gene targets: p6 line 23: why 418 genes out of 554?
 - o Suppl. Data 6 seems to have 519 unique genes – what are they? They are neither all 554 cleaved targets, nor the 418 from the GO analysis (but do include some TE's)?
- How were the phasiRNA targets predicted, and how many were found? This is of particular interest since many previous studies did not (or not confidentially) find targets. Please detail the method and parameters.
- Regarding cleavage of target sites: Initially, I was not clear about the following points, please add them into the results/figure legends:
 - o Is the specific cleavage of target sites better than random? (Yes, but criteria for selection above 10% of total mRNA only found in methods)
 - o Figure: Red line is explained but also mention the red dot at the end that is the actual data point, I overlooked it at first and got confused.
- In the end, only a small proportion (~1/10) of the cleaved target genes showed de-repression in mel1 and osrdr6 mutant meiocytes.
 - o I thus wonder what the functionality of them then is and if this is relevant at all, since phasiRNA-mediated cleavage might not affect the mRNA levels in most cases.
 - o Among the targets was RAD51, an important gene for meiotic recombination, but it seems it was excluded from the set of genes for GO analysis and expression analysis. Mentioning it will connect this study to a meiotic key player.

- Many of the genes (carbohydrate metabolism, Rad51) are just generally high expressed in early meiocytes and thus also their degradation products – how real or relevant is the detected cleavage then?

o Mostly low fold-change of target gene expression in mutants

o How big of a pool was there for the phasiRNA-predicted genes?

- GO analysis can be skewed dependent on the quality of the annotation. I would suggest to try GO analysis for putative Arabidopsis homologs of the 554 target genes.

Other points:

- I would suggest to emphasize that it makes sense that phasiRNAs from whole phasi loci are decreased in *rdr6* mutants while only specific, individual phasiRNAs are decreased in *mel1* mutants (general biosynthesis via specific stabilization in pathway).

- In the abstract, phasiRNAs are claimed “essential for male sterility” – this is not the case but true under different environmental conditions (see rice PGMS and maize *dcl5* mutant).

- Page 5 line 9/10 “The abundances of 24-nt phasiRNAs were also reduced albeit to a lesser extent”. This might be due to the sample stage (prophase I) whereas more differences might have been found at later stages.

Minor comments:

(additionally see attached files with comments)

- Suppl. Data S2. Typo at very end of tables: “is caculated” ◊ “is calculated”; also maybe put the table information at the top of the big tables, not underneath. Also, it is helpful for the reader, if big tables always display the upper column titles. For this, click on the row underneath the table titles, choose the “View” Excel tab, and select “Freeze Panes”.

Reviewer #2 (Remarks to the Author):

The manuscript submitted by Jiang and colleagues focus on the identification and characterization of transcripts that are putative targets of a group of phasiRNAs known for being specifically expressed in rice inflorescence and mostly triggered by miR2118 and miR2775.

In the first part of this work, the authors analyze the expression profile of those phasiRNAs in different stages and tissues related to male germ cells. Their analyses show that 21 nt long phasiRNAs are mainly expressed in the earlier stages of germ cell development and the 24 nt long phasiRNA are more abundant later on. Despite providing enrichment and more detail to our knowledge about those molecules, the results presented here are not entirely new (Fei et al 2016, doi:10.1093/jxb/erw361 and Komiya 2017, DOI 10.1007/s10265-016-0878-0) and have more a character of preparing and confirming the libraries for following analyses.

They also describe the isolation of *rdr6* and *mel1* mutants, which are used for phasiRNA profile and gene expression analysis later on.

Nonetheless, the focus of the work and the most relevant results come from the identification and characterization of putative targets of those phasiRNAs. Given the importance of rice and the involvement of this pathway with the plant fertility, the results presented here are of special interest. These findings have an additional relevance due to the fact that prior attempts have failed to find such targets (Song et al 2011, doi: 10.1111/j.1365-313X.2011.04805.x and Komiya et al 2014, doi: 10.1111/tpj.12483). However, it is my opinion that the characterization of such targets is quite superficial, to not say poor, especially taking in consideration the level of publication wanted. I have therefore included below a few suggestions, which I believe could help to improve the impact of this work.

- Regarding the 572 21 nt long phasiRNAs with putative targets, from how many PHAS loci are they coming from? Is there one PHAS gene with more relevant contribution?

- The same regarding the genes targeted by phasiRNAs and de-repressed (≥ 1.5 fold change, page

7, line 4). What are these genes, in which path are they involved. What are the phasiRNAs that target them. Are they coming from multiple PHAS loci or from just a few?

- Also, given the high number of target genes that do not show variation in expression (at least not above the cut off, page 7, line 8) it would be important to include a paragraph in discussion about it.

- Since mel1 and rdr6 mutants are available, it would be very interesting to have the degradome from these lines done. With this information it would be possible to not only confirm the degradome results obtained so far, but also to know if the targeting events are MEL1-dependent or dependent of another AGO.

In addition, the characterization of *osrdr6-2* is very poor and needs to be improved. Differently from *mel1-4*, *osrdr6-2* harbors a point mutation and its effect on the protein activity is unclear. This can have serious consequences on the interpretation of some experiments. For instance, the authors found that around 20% of the phasiRNAs are RDR6-independent. This number seems a bit high, for what is known about secondary siRNAs. One possible explanation would be that *osrdr6-2* is a knock-down, with part of the enzyme activity still present and therefore, not affecting all phasiRNAs. It would be important to add a few results, for instance, is TAS3-derived tasiRNAs completely gone? Where is the mutation in the RDR6 gene. Does it affect an important domain?

Minor comments:

- The authors should carefully review the manuscript to look for typos and grammar mistakes. For instance: Page 3, line 11 – “cleavage of cleave”; page 3, line 19 – “stages,,”; Supp figure 6 – “terad”

- In figure 3a (and the similar ones in supp figures) the presence of the red line is confusing. It took me quite a long time to realize the red dot on top of the line, which made me believe that amount of counts were much less than is indicated.

Reviewer #3 (Remarks to the Author):

This manuscript describes analysis of phasiRNAs in rice and attempts to demonstrate their mode of action. The paper includes several sets of novel sequencing data, but the conclusions drawn are either primarily confirmatory or are insufficiently supported.

First, sRNAs from dissected tissues were sequenced to identify phasiRNAs in the male germ line (wild type, *osrdr6*, and *mel1*). These data are consistent with previously published analysis in rice and other systems (Fei 2016 J Exp Bot, Zhai 2015 PNAS, Song 2012 Plant J, and Komiyama 2014 Plant J). Therefore, while the datasets are novel (and could be useful to the community for other purposes, eg identifying tissue-specific miRNAs), the conclusions are confirmatory.

The manuscript also includes annotation of phasiRNA loci that were not previously identified, however I am concerned that the parameters used for this analysis were overly permissive. The PhaseTank algorithm is reported to return a phase score per locus. What phase score threshold was used and how many of the annotated phasiRNA loci met this threshold? Alternatively, there are other programs capable of detecting phasing (see list in Morgado and Johannes 2019 Briefings in Bioinf), and overlap between these programs could be used to add robustness to the phasiRNA categorization. This manuscript should also address the issue that abundantly-expressed 24-nt loci can have elevated phasing scores (Polydore et al, Plant Direct 2018). Misannotation of these loci as phasiRNAs when they are actually Pol IV-derived siRNAs would explain why so many of them are not targeted by miR2275 and do not require *osrdr6*.

Next the manuscript includes high-sensitivity degradome sequencing from dissected rice tissues. Regarding the method: description of this approach will be of broad interest, however there is no

benchmarking of this method compared to traditional degradome sequencing. This lack of comparison makes it difficult to trust that the weak signals reported are not artifacts of the process.

The degradome signals shown in Figure 3a are convincing, but many of those shown in Figure S5 are not. In many cases, sequence tags are as frequent at non-target sites as at the predicted target site. For example, from the first page of 20 hits, I count 5 examples that are not convincing: 4_510, 7_459, 21_457, 21_520, 28_142. For many cases there are very few sequence tags, increasing the likelihood that these are false positives. The methods state that the predicted cleavage site needs to be only $\geq 10\%$ of the tags matching to a transcript, which is low. I am also concerned that the "mispair score" (is this the penalty score that psRobot outputs?) is very high and that many of the phasiRNA-target predictions are weak. With liberal thresholds to define phasiRNAs, to predict phasiRNA-mRNA pairing, and to identify degradome tags, it is not surprising that many genes were found. Some metrics to show that these cleavage events can be found with stricter thresholds are needed.

In addition to suggesting that mRNA cleavage is occurring, the manuscript concludes that this is more abundant in prophase I samples (Figure 3c). Yet the legend suggests that the read counts in the heat map are not normalized to total library size. There are over 90 million mapped reads for prophase I meiocytes, but less than 20 million reads for tetrads (~50 million reads for microspores). Given that the majority of targets have appropriate tags 4-5 sequence tags overall, it is likely that the tetrad and microspore libraries weren't sequenced sufficiently to detect cleavage products that were found in meiocytes, and therefore no conclusion can be drawn about their absence.

As additional evidence to show whether the putative phasiRNA targets are repressed by phasiRNAs, the paper includes RNAseq from *rdr6* and *mel1* samples. It is a stretch to conclude that because 45/28 genes were slightly upregulated in *rdr6* and *mel1* that they are normally repressed by phasiRNAs. These mutants have defective meiocytes and therefore some level of differential gene expression is to be expected. A better question is whether predicted phasiRNA target were enriched among the differentially expressed genes. In other words, given X expressed genes, Y differentially expressed genes, and Z predicted targets, is Z/Y higher than Z/X?

Finally, the authors conclude that carbohydrate synthesis is being modulated based on GO-term enrichment of putative phasiRNA targets, however GO term enrichment is based on the number of genes with a given ontology that could be present in the sample (the null distribution). Given that very specific cell types are being tested and they are likely to have a different distribution of expressed genes, the authors should check whether these GO terms are enriched in microspores generally (and therefore enriched in phasiRNA targets within microspores). GO term enrichment is weak data at the best of times, but without analysis of the expressed transcripts in the tissue types, it is particularly tenuous.

Throughout, the manuscript is missing information about how replicates were handled for all sequence types and also whether these replicates arise from independent samples (biological replicates rather than technical replicates). It does not appear that variation between replicates was used in the sRNA analysis. For example, the methods state that "read counts of each phasiRNA from wild-type and mutant plants were compared using a Fisher's exact test. The resulting P value was adjusted to estimate false discovery rate." A method that uses replicate data (eg, DESeq) is needed to make conclusions regarding change in expression.

Similarly, for degradome sequencing, it appears that replicates were merged and treated as a single sample. Identification of sequence tags at the predicted site in multiple replicates would go a long way toward convincing this reviewer that mRNAs are targeted by phasiRNAs.

Finally, for RNAseq data, please be explicit about whether replicate data was entered into Cuffdiff,

or whether replicates were merged and assessed as single datasets.

Minor comments:

Pg 5, line 23: Just because an AGO contains a conserved catalytic triad (more accurately catalytic tetrad - see Sheng 2014 PBAS), doesn't mean they cleave target mRNAs. Examples include AGO4 from plants; and Twi1p from Tetrahymena. "Slicing" activity could just be a mechanism of AGO loading.

It is not clear exactly what is shown in Figure 4c. It appears to be FPKM from the sequencing, but there is no description of the error bars. Given that there must be 3 data points for each bar, it is more appropriate to show these datapoints rather than an average.

Although the paper is generally very well written and clear, there are a few typos throughout. Eg, Line 11: "direct cleavage of cleave hundreds of target genes through mRNA cleavage"

We appreciate the constructive comments made by the reviewers. We have provided additional data and revised our manuscript to address the concerns raised by the reviewers. We wish the revisions are sufficient and the manuscript is now acceptable for publication. Point-by-point responses are listed below.

Reviewer #1:

Jiang et al. make an important step forward to understanding the biological functions of reproductive phasiRNAs. By using isolated meiocytes from rice and integrating sRNA-seq with mRNAseq and degradome sequencing adopted for low input material, they identified cleavage of specific targets in early prophase I, mediated by 21nt phasiRNAs, namely 554 protein coding genes and 102 transposons. This is of note since most previous studies in maize and rice did not find target genes. An additional interesting finding is that no target gene cleavage was found for 24nt phasiRNAs which might point to diverse functions of those otherwise similar classes of reproductive phasiRNAs. The manuscript is well written and precise, figures support the drawn conclusions, and the data is sufficient for the claims made. Addressing my questions and comments below will help to get to a final version that is clear and connected with relevant other knowledge. The study is of high interest to the field of plant reproductive phasiRNA research. However, the manuscript would benefit from connecting the findings with previous studies to put them into contexts:

Tamim et al 2018. Cis-directed cleavage and nonstoichiometric abundances of 21-nucleotide reproductive phased small interfering RNAs in grasses: This paper also found that specific phasiRNAs are more stable than others. Are those related with the ones mediating cleavage? Same question for the results from the MEL1 paper cited and the current *mell1*-data – are the MEL1-stabilized phasiRNAs overlapping the ones mediating cleavage?

Response: Thanks for the suggestion. To examine whether phasiRNAs mediating cleavage are more stable than others, we compared the abundances of phasiRNAs with targets and those without targets in prophase I meiocytes. The results showed that phasiRNAs with cleavage targets were more abundant than others (Supplementary Fig. 9a). PhasiRNAs with cleavage targets were decreased to a greater extent than the ones without cleavage targets in abundance (Supplementary Fig. 9b) in the *mell1-4* mutant. We also performed analysis using the published MEL1-IP sRNA-seq data ¹. We found that 71% of phasiRNAs with targets were

MEL1-associated, while a lower percentage (50%) of phasiRNAs without targets were MEL1-associated. Furthermore, phasiRNAs with cleavage targets were more abundant in MEL1 (Supplementary Fig. 9c, d). These findings suggest that phasiRNAs mediating cleavage are more stable, presumably because of their association with MEL1. We have included these new data in the manuscript.

Fan et al 2016. *PMS1T*, producing phased small-interfering RNAs, regulates photoperiod-sensitive male sterility in rice; and Wang et al 2011. Comparative transcriptomes profiling of photoperiod-sensitive male sterile rice nongken 58S during the male sterility transition between short-day and long-day: These papers describe an important rice 21phasi locus that confers conditional male sterility, and identify target genes or transcriptional changes. Does the present dataset include this finding and/or can confirm or extend it?

Response: *PMS1T* in rice Nongken 58S contains a point mutation nearby the miR2118 recognition site, which makes *PMS1T* produce abundant phasiRNAs under long-day conditions. Low levels of phasiRNAs can be generated from *PMS1T* without the point mutation regardless of day length². *PMS1T* does not contain the point mutation in rice Nipponbare, 3037 and HHZ. Despite this, phasiRNAs generated from *PMS1T* were detected in these rice lines by our sRNA sequencing (Supplementary Fig. 1c) and one target (*LOC_os03g31410*, expressed genes) was detected by our degradome sequencing (Supplementary Fig. 5). This suggests that our sRNA and degradome sequencing are sensitive and reliable.

- Put into wider context: very similar to animal piRNAs (silencing of meiotic genes and TEs): Goh et al 2015. piRNA-directed cleavage of meiotic transcripts regulates spermatogenesis

Response: As suggested, we have discussed the similarity between phasiRNAs and animal piRNAs.

- Numbers that seem not to match to me: Only about half of the 24nt phasi loci are targeted by miR2775 but 94.5% had miR2275-directed cleavage patterns?

Response: Sorry about the confusion. Half is the percentage of predicted miR2275-targeting loci in all 24-nt *PHAS* loci. 94.6% (95.3% in the revised manuscript) is the percentage of loci with miR2275-directed cleavage signals in all predicted miR2275-targeting loci.

Concerning protein coding gene targets: p6 line 23: why 418 genes out of 554?

Response: In total, 554 protein-coding genes were identified as 21-nt

phasiRNA-guided cleavage targets in early prophase I meiocytes, tetrads and microspores. 418 is the number of 21-nt phasiRNA-guided cleavage targets (protein-coding genes) in early prophase I meiocytes. We could only collect early prophase I meiocytes from mutants and perform RNA-seq analysis to examine differential expression of these 418 genes. In the revised manuscript, the numbers 554 and 418 have been changed to 367 and 253 because we applied more stringent methods and criteria for target selection.

Suppl. Data 6 seems to have 519 unique genes – what are they? They are neither all 554 cleaved targets, nor the 418 from the GO analysis (but do include some TE's)

Response: In total, 554 protein-coding genes and 102 TEs were identified as 21-nt phasiRNA-guided cleavage targets in early prophase I meiocytes, tetrads and microspores. 519 unique genes are 418 protein-coding genes and 101 TEs that were identified as 21-nt phasiRNA-guided cleavage targets in early prophase I meiocytes. These numbers have been changed in the revised manuscript because we applied more stringent methods and criteria for target selection.

- How were the phasiRNA targets predicted, and how many were found? This is of particular interest since many previous studies did not (or not confidentially) find targets. Please detail the method and parameters.

Response: All of annotated phasiRNAs were aligned to rice cDNA sequences downloaded from rice genome database (MSU v7) using psRobot_tar (-ts 3.5 -fp 2 -tp 17)³. Mispair score was calculated by psRobot using the following rules: 1) Mismatches, gaps or bulges incur a penalty of +1; 2) G:U pairs incur a penalty of +0.5; 3) Penalty scores outside the core base-paired region (2-17) are reduced by half. Genes (including TEs) that pair with phasiRNAs with mispair scores ≤ 3.5 were selected as an initial pool of potential phasiRNA targets. The 5' ends of degradome tags derived from phasiRNA cleavage fragments should precisely match the 10th nucleotide of phasiRNA complementary sites. Such tags were searched in the degradome libraries using psRobot_deg³. All hit transcripts are categorized based on the abundance of the diagnostic cleavage tag relative to the overall profile of degradome tags matching the target essentially as previously described^{4,5}. Category 4, 1 degradome tag at the cleavage site; Category 3, degradome tags at the cleavage site > 1 , but \leq the average number of degradome tags on the transcript; Category 2, degradome tags at the cleavage site > 1 , above the average but not the maximum on the transcript; Category 1, degradome tags at the cleavage site > 1 , equal to the

maximum on the transcript, but there are > 1 sites at maximum value; Category 0, > 1 degradome tags, equal to the maximum on the transcript, and there is only 1 site at the maximum value. Category 0 or Category 1 transcripts that have ≥ 4 degradome tags at the cleavage site, which account for $\geq 10\%$ of the total degradome tags along the transcript, were annotated as phasiRNA targets.

Because MEL1 is specifically expressed in germ cells prior to meiosis, phasiRNAs should direct target mRNA cleavage in these cells. We believe that other studies did not find targets because they didn't collect a pure population of premeiotic germ cells or germ cells in meiotic prophase for degradome sequencing. There are high levels of background degradation signals when using spikelets for degradome sequencing. It is not easy to distinguish phasiRNA-guided cleavage signals from background degradation signals. We have detailed the method and parameters.

Is the specific cleavage of target sites better than random? (Yes, but criteria for selection above 10% of total mRNA only found in methods)

Response: We previously used less-stringent criteria in order to identify more candidates for future investigation. In the revised manuscript, we have used more stringent criteria as elaborated above.

Figure: Red line is explained but also mention the red dot at the end that is the actual data point, I overlooked it at first and got confused.

Response: We have deleted the red line. We have revised our figure legend to make it clear that the red dot represents degradome tag at +1 position of 21-nt phasiRNA-guided cleavage site.

In the end, only a small proportion ($\sim 1/10$) of the cleaved target genes showed de-repression in *mell1* and *osrd6* mutant meiocytes. I thus wonder what the functionality of them then is and if this is relevant at all, since phasiRNA-mediated cleavage might not affect the mRNA levels in most cases.

Response: First, in gene regulatory networks, there are many feedback loops. The genes that are posttranscriptionally derepressed could be transcriptionally repressed to maintain gene expression homeostasis. Second, phasiRNAs may regulate target gene expression using other modes of action. Such regulation may affect gene derepression in mutants. Third, we used a cutoff (≥ 1.5 fold change) to select the derepressed targets. Genes, which had significantly increase of expression (q-value ≤ 0.05) but did not pass this cutoff, were not considered as derepressed targets (Fig. 4c,

Supplementary Data 7). For all these reasons, the percentage of genes showing derepression in the mutants could be underestimated.

Among the targets was RAD51, an important gene for meiotic recombination, but it seems it was excluded from the set of genes for GO analysis and expression analysis. Mentioning it will connect this study to a meiotic key player.

Response: Thanks for pointing out this. We have mentioned this in the revised manuscript.

Many of the genes (carbohydrate metabolism, Rad51) are just generally high expressed in early meiocytes and thus also their degradation products – how real or relevant is the detected cleavage then?

Response: We have used stringent criteria for phasiRNA target identification. We only detected cleavage mediated by 21-nt phasiRNAs. Using the same pipeline, we could not identify 24-nt phasiRNA-guided cleavage targets, suggesting that the detected cleavage is real. To further investigate the relationship between gene expression level and the chance of cleavage detection, we calculated the Pearson correlation coefficient (p.c.c) between expression levels of targets and degradome tags at cleavage sites (Fig. 1a in this response). A correlation coefficient of 0.16 indicates a very weak correlation, suggesting that the detected cleavage is real and does not only come from highly expressed genes. Moreover, we performed the permutation test 3 times to determine whether a higher percentage of phasiRNA targets could be identified from a defined set (n=1000) of randomly selected genes, which had genes expression levels comparable to our identified targets, than from all genes in the genome (Fig. 1b in this response). The result shows that there is no difference between the percentage of phasiRNA targets in randomly selected genes and the expected percentage, further suggesting that the chance to be identified as phasiRNA targets is minimally affected by gene expression level.

Fig. 1. Relationship between gene expression level and the chance of cleavage detection. **a**, Relative expression abundance and degradome tags of target genes. **b**, permutation test.

Mostly low fold-change of target gene expression in mutants

Response: First, in gene regulatory networks, there are many feedback loops. The genes that are posttranscriptionally derepressed could be transcriptionally repressed to maintain gene expression homeostasis. Second, phasiRNAs may regulate target gene expression using other modes of action. Such regulation may affect gene derepression in mutants. Thus, the fold-change of target gene expression could be underestimated. On the other hand, the importance of phasiRNAs should lie in their capability to target many genes. They do not necessarily induce high fold-changes of target gene expression. Thus, mutants often have low fold-changes of target gene expression.

How big of a pool was there for the phasiRNA-predicted genes?

Response: Genes (including TEs) that pair with phasiRNAs with mispair scores ≤ 3.5 were selected as an initial pool of phasiRNA targets. 55534 genes were predicted to be targeted by 11803 21-nt phasiRNAs. Cleavage products of these 55534 predicted phasiRNA targets were then searched in degradome libraries using psRobot_deg to identify high-confidence targets.

GO analysis can be skewed dependent on the quality of the annotation. I would suggest to try GO analysis for putative Arabidopsis homologs of the 554 target genes.

Response: Our GO analysis of the 418 (253 in the revised manuscript) rice protein-coding genes was based on GO analysis of *Arabidopsis* homologs.

Other points:

I would suggest to emphasize that it makes sense that phasiRNAs from whole phasi loci are decreased in *rdr6* mutants while only specific, individual phasiRNAs are decreased in *me11* mutants (general biosynthesis via specific stabilization in pathway).

Response: We have added this in the revised manuscript.

In the abstract, phasiRNAs are claimed “essential for male sterility” – this is not the case but true under different environmental conditions (see rice PGMS and maize *dcl5* mutant).

Response: We have revised this sentence to make it accurate.

Page 5 line 9/10 “The abundances of 24-nt phasiRNAs were also reduced albeit to a lesser extent”. This might be due to the sample stage (prophase I) whereas more differences might have been found at later stages.

Response: We agree with the reviewer that a larger abundance difference of 24-nt phasiRNAs could be found at later stages. However, the development of male germ cells in *osrdr6* and *mell1* mutants stop at meiotic prophase I. We are unable to collect the cells at later stages from the mutants for the analysis.

Minor comments: (additionally see attached files with comments) Suppl. Data S2. Typo at very end of tables: “is caculated” □ “is calculated”; also, maybe put the table information at the top of the big tables, not underneath. Also, it is helpful for the reader, if big tables always display the upper column titles. For this, click on the row underneath the table titles, choose the “View” Excel tab, and select “Freeze Panes”.

Response: We have corrected the mistake and revised the tables as suggested.

Reviewer #2:

The manuscript submitted by Jiang and colleagues focus on the identification and characterization of transcripts that are putative targets of a group of phasiRNAs known for being specifically expressed in rice inflorescence and mostly triggered by miR2118 and miR2775. In the first part of this work, the authors analyze the expression profile of those phasiRNAs in different stages and tissues related to male germ cells. Their analyses show that 21 nt long phasiRNAs are mainly expressed in the earlier stages of germ cell development and the 24 nt long phasiRNA are more abundant later on. Despite providing enrichment and more detail to our knowledge about those molecules, the results presented here are not entirely new (Fei et al 2016, doi:10.1093/jxb/erw361 and Komiya 2017, DOI 10.1007/s10265-016-0878-0) and have more a character of preparing and confirming the libraries for following analyses. They also describe the isolation of *rdr6* and *mell1* mutants, which are used for phasiRNA profile and gene expression analysis later on. Nonetheless, the focus of the work and the most relevant results come from the identification and characterization of putative targets of those phasiRNAs. Given the importance of rice and the involvement of this pathway with the plant fertility, the results presented here are of special interest. These findings have an additional relevance due to the fact that prior attempts have failed to find such targets (Song et al 2011, doi: 10.1111/j.1365-313X.2011.04805.x and Komiya et al 2014, doi: 10.1111/tpj.12483). However, it is my opinion that the characterization of such targets is quite superficial, to not say poor, especially taking in consideration the level of publication wanted. I have therefore included bellow a few suggestions, which I believe could help to

improve the impact of this work.

Response: Thanks for the comments. We respectfully disagree that our analyses of the phasiRNA targets are superficial. We would like to point out that this is the first time to identify phasiRNA targets and to show phasiRNAs direct target cleavage.

- Regarding the 572 21 nt long phasiRNAs with putative targets, from how many PHAS loci are they coming from? Is there one PHAS gene with more relevant contribution?

Response: These 572 21-nt phasiRNAs are from 501 *PHAS* loci. 77.24% of these *PHAS* loci have one target gene, 16.37% of these *PHAS* loci have two target genes, and 6.39% of these *PHAS* loci have three or more target genes (Fig. 2 in this response).

Fig. 2 Pie chart showing the percentages of *PHAS* loci with 1, 2 and ≥ 3 target genes.

The same regarding the genes targeted by phasiRNAs and de-repressed (≥ 1.5 -fold change, page 7, line 4). What are these genes, in which path are they involved? What are the phasiRNAs that target them. Are they coming from multiple *PHAS* loci or from just a few?

Response: We have performed pathway enrichment analysis of genes targeted by phasiRNAs and de-repressed in *mell-4* or *osrdr6-2*. However, the analysis resulted in no enriched pathways. More detailed information about these genes and phasiRNAs targeting them have been provided in Supplementary Data 7. These phasiRNAs are derived from multiple *PHAS* loci.

Also, given the high number of target genes that do not show variation in expression (at least not above the cut off, page 7, line 8) it would be important to include a paragraph in discussion about it.

Response: First, in gene regulatory networks, there are many feedback loops. The genes that are posttranscriptionally derepressed could be transcriptionally repressed to maintain gene expression homeostasis. Second, phasiRNAs may regulate target gene expression using other modes of action. Such regulation may affect gene derepression in mutants. For these and other reasons, many genes may not show big variation in expression. We have included a paragraph to discuss this in our revised manuscript.

Since *mell1* and *rdr6* mutants are available, it would be very interesting to have the degradome from these lines done. With this information it would be possible to not only confirm the degradome results obtained so far, but also to know if the targeting events are MEL1-dependent or dependent of another AGO.

Response: Thanks for the suggestion. We have performed degradome sequencing of *mell1-4* and *osrdr6-2* mutants and their corresponding wild-type lines and provided the results. Among 316 genes (253 protein coding genes and 63 TEs) targeted by 21-nt phasiRNAs in early prophase I meiocytes of Nipponbare, 131 (121 protein-coding genes and 10 TEs) and 133 genes (121 protein-coding genes and 12 TEs) were detected to be 21-nt phasiRNA targets in early prophase I meiocytes of 3037 and HHZ, respectively. For over 60% of targets, cleavage events at phasiRNA-guided cleavage site were markedly reduced in *osrdr6-2* and *mell1-4* (Supplementary Fig. 8, Supplementary Data 6), indicating that most cleavage events are dependent on MEL1 and OsRDR6.

In addition, the characterization of *osrdr6-2* is very poor and needs to be improved. Differently from *mell1-4*, *osrdr6-2* harbors a point mutation and its effect on the protein activity is unclear. This can have serious consequences on the interpretation of some experiments. For instance, the authors found that around 20% of the phasiRNAs are RDR6-independent. This number seems a bit high, for what is known about secondary siRNAs. One possible explanation would be that *osrdr6-2* is a knock-down, with part of the enzyme activity still present and therefore, not affecting all phasiRNAs. It would be important to add a few results, for instance, are TAS3-derived tasiRNAs completely gone? Where is the mutation in the RDR6 gene? Does it affect an important domain?

Response: The *osrdr6* knockout mutation is lethal. The *osrdr6-2* mutant we used is a weak allele. It harbors a point mutation in the RDRP domain of OsRDR6. The amino acid Arginine (R), which is mutated to Leucine (L) in *osrdr6-2*, is highly conserved in eukaryotes. We have added the domain information of RDR6 and alignment of RDR6

partial protein sequence in our revised manuscript (Supplementary Fig. 2). As suggested, we detected the levels of TAS3-derived tasiRNAs to assess the level of residual RDR6 activity in *osrdr6-2*. The rice genome has four *TAS3* loci. tasiRNAs

derived from *TAS3a1* remained unaltered in abundance in *osrdr6-2*. tasiRNAs derived from *TAS3a2* were hardly detected in wild-type 3037 and *osrdr6-2*. tasiRNAs derived from *TAS3b1* and *TAS3b2* were reduced in abundance but not eliminated in *osrdr6-2* (Fig. 3 in this response). These data suggest that there is a substantial amount of residual RDR6 activity in *osrdr6-2*. This could lead to an overestimation of the percentage.

Fig. 4 Abundance of TAS3-derived tasiRNAs in 3037 and *osrdr6-2*.

Minor comments:

The authors should carefully review the manuscript to look for typos and grammar mistakes. For instance: Page 3, line 11 – “cleavage of cleave”; page 3, line 19 – “stages.”; Supp figure 6 – “terad”

Response: We have double-checked our manuscript and corrected grammar mistakes and typos.

In figure 3a (and the similar ones in supp figures) the presence of the red line is confusing. It took me quite a long time to realize the red dot on top of the line, which make me believe that amount of counts were much less that is indicated.

Response: We have deleted the red line. We have revised our figure legend to make it clear that the red dot represents degradome tag at +1 position of 21-nt phasiRNA-guided cleavage site.

Reviewer #3

This manuscript describes analysis of phasiRNAs in rice and attempts to demonstrate their mode of action. The paper includes a several sets of novel sequencing data, but the conclusions draw are either primarily confirmatory or are insufficiently supported.

First, sRNAs from dissected tissues were sequenced to identify phasiRNAs in the male germ line (wild type, *osrdr6*, and *mel1*). These data are consistent with previously published analysis in rice and other systems (Fei 2016 J Exp Bot, Zhai 2015 PNAS, Song 2012 Plant J, and Komiya 2014 Plant J). Therefore, while the datasets are novel (and could be useful to the community for other purposes, eg identifying tissue-specific miRNAs), the conclusions are confirmatory.

The manuscript also includes annotation of phasiRNA loci that were not previously identified, however I am concerned that the parameters used for this analysis were overly permissive. The PhaseTank algorithm is reported to return a phase score per locus. What phase score threshold was used and how many of the annotated phasiRNA loci met this threshold? Alternatively, there are other programs capable of detecting phasing (see list in Morgado and Johannes 2019 Briefings in Bioinf), and overlap between these programs could be used to add robustness to the phasiRNA categorization.

Response: As suggested, we have used two programs PhaseTank and Unitas to identify *PHAS* loci. We found that 92.4% of 21-nt *PHAS* loci and 87.83% of 24-nt *PHAS* loci identified by PhaseTank and 80.88% of 21-nt *PHAS* loci and 19.17% of 24-nt *PHAS* loci identified by Unitas overlapped. *PHAS* loci in our final lists were those identified by both Phase and Unitas. In total, 11,810 21-nt phasiRNAs and 1,689 24-nt phasiRNAs generated from 2173 loci and 195 loci, respectively, were identified (Fig. 1b and Supplementary Data 2, 3). Among them, 899 21-nt phasiRNAs and 178 24-nt phasiRNAs from 317 loci and 58 loci, respectively, have not been previously annotated (Supplementary Data 2, 3). These loci were identified as *PHAS* loci, not because of overly permissive parameters. These loci met the criterion that has been used by others for phasiRNA identification, which is phase score > 10. The phase score was calculated by applying our method to data obtained by Fei et al., 2016⁶.

This manuscript should also address the issue that abundantly-expressed 24-nt loci can have elevated phasing scores (Polydore et al, Plant Direct 2018). Misannotation of these loci as phasiRNAs when they are actually Pol IV-derived siRNAs would explain why so many of them are not targeted by miR2275 and do not require *osrdr6*.

Response: We agree with the reviewer that 24-nt Pol IV- and RDR2-dependent siRNA loci may have elevated phasing scores. We thus carefully examined 24-nt *PHAS* loci that are not targeted by miR2275. We found that, first, most of these 24-nt sRNA loci are not located in transposon and repeat regions and they exhibit distinct

phased sRNA patterns (Fig. 4 in this response). Second, these 24-nt sRNAs, like miR2275-triggered 24-nt phasiRNAs, are highly expressed in male meiocytes, but not in somatic cells (Fig. 4 in this response). Third, precursors of these 24-nt sRNA loci produce strong signals in prophase I meiocytes' mRNA-Seq data but not in embryo's mRNA-Seq data, indicating that they are transcribed by Pol II but not Pol IV (Fig. 4

in this response). For these reasons, we conclude that these loci are 24-nt *PHAS* loci but not Pol IV- and RDR2-dependent siRNA loci. Many of 24-nt phasiRNAs appeared to be RDR6-independent. A recent genome-wide study of reproductive phasiRNAs in *Asparagus* and other monocots also detected many miR2275- and RDR6-independent 24-nt phasiRNAs derived from inverted repeats, suggesting that their precursors may form foldback and then be processed by Dicer, bypassing the need for an RDR protein⁷.

Fig. 4 Genome browser view of 24-nt *PHAS* loci not targeted by miR2275.

The manuscript includes high-sensitivity degradome sequencing from dissected rice tissues. Regarding the method: description of this approach will be of broad interest,

however there is no benchmarking of this method compared to traditional degradome sequencing. This lack of comparison makes it difficult to trust that the weak signals reported are not artifacts of the process.

Response: We have added a detailed protocol in the Methods. The traditional PARE method needs large amount of total RNA. It is impossible to isolate large amount of RNA from pure populations of male germ cells. Thus, we are unable to compare our results with those obtained from traditional degradome sequencing. However, we have used stringent criteria for target prediction in our revised manuscript to avoid false positives. In our revised manuscript, 1883 21-nt *PHAS* loci and 89 24-nt *PHAS* loci were predicted to be targeted by miR2118 and miR2275 respectively. 74.8% of miR2118-triggered *PHAS* loci and 95.3% of miR2275-triggered *PHAS* loci were detected to have miR2118- and miR2275-directed cleavage signals, indicating that our degradome sequencing is of high quality. For 24-nt phasiRNAs, 19714 genes were predicted as their targets. However, no genes were identified as their cleavage targets. These predicted 24-nt phasiRNA targets could serve as a negative control to verify that signals at 21-nt phasiRNA targets are not artifacts.

The degradome signals shown in Figure 3a are convincing, but many of those shown in Figure S5 are not. In many cases, sequence tags are as frequent at non-target sites as at the predicted target site. For example, from the first page of 20 hits, I count 5 examples that are not convincing: 4_510, 7_459, 21_457, 21_520, 28_142. For many cases there are very few sequence tags, increasing the likelihood that these are false positives. The methods state that the predicted cleavage site needs to be only $\geq 10\%$ of the tags matching to a transcript, which is low. I am also concerned that the “mispair score” (is this the penalty score that psRobot outputs?) is very high and that many of the phasiRNA-target predictions are weak. With liberal thresholds to define phasiRNAs, to predict phasiRNA-mRNA pairing, and to identify degradome tags, it is not surprising that many genes were found. Some metrics to show that these cleavage events can be found with stricter thresholds are needed.

Response: We previously used less-stringent criteria in order to identify more candidates for future investigation. In the revised manuscript, we have used more stringent criteria as elaborated in our response to Reviewer #1.

In addition to suggesting that mRNA cleavage is occurring, the manuscript concludes that this is more abundant in prophase I samples (Figure 3c). Yet the legend suggests that the read counts in the heat map are not normalized to total library size. There are

over 90 million mapped reads for prophase I meiocytes, but less than 20 million reads for tetrads (~50 million reads for microspores). Given that the majority of targets have appropriate tags 4-5 sequence tags overall, it is likely that the tetrad and microspore libraries weren't sequenced sufficiently to detect cleavage products that were found in meiocytes, and therefore no conclusion can be drawn about their absence.

Response: We thank the reviewer for raising this point. We have replaced raw counts with reads normalized against genome-matched reads (RP10M). The conclusion remains the same.

As additional evidence to show whether the putative phasiRNA targets are repressed by phasiRNAs, the paper includes RNA seq from *rdr6* and *mel1* samples. It is a stretch to conclude that because 45/28 genes were slightly upregulated in *rdr6* and *mel1* that they are normally repressed by phasiRNAs. These mutants have defective meiocytes and therefore some level of differential gene expression is to be expected. A better question is whether predicted phasiRNA target were enriched among the differentially expressed genes. In other words, given X expressed genes, Y differentially expressed genes, and Z predicted targets, is Z/Y higher than Z/X?

Response: We have compared the percentage of significantly up-regulated phasiRNA cleavage targets (P value ≤ 0.05 , Fisher Exact Test) in total phasiRNA cleavage targets with percentage of significantly up-regulated genes in total expressed genes. As expected, phasiRNA targets are significantly enriched among up-regulated genes (Fig. 5 in this response).

Fig. 5 Enrichment analysis of phasiRNA targets among differentially expressed genes. The percentages of significantly up-regulated phasiRNA cleavage targets (P value ≤ 0.05 , Fisher Exact Test) in total phasiRNA cleavage targets were compared with the percentages of significantly up-regulated genes in total expressed genes.

Finally, the authors conclude that carbohydrate synthesis is being modulated based on GO-term enrichment of putative phasiRNA targets, however GO term enrichment is based on the number of genes with a given ontology that could be present in the sample (the null distribution). Given that very specific cell types are being tested and they are likely to have a different distribution of expressed genes, the authors should check whether these GO terms are enriched in microspores generally (and therefore enriched in phasiRNA targets within microspores). GO term enrichment is weak data at the best of times, but without analysis of the expressed transcripts in the tissue types, it is particularly tenuous.

Response: We have performed GO term enrichment of phasiRNA targets in early prophase I meiocytes using all expressed genes in early prophase I meiocytes, but not all of the annotated protein-coding genes in rice, as a background control (the null distribution). GO term frequencies in phasiRNA targets were compared to GO term frequencies in expressed genes in early prophase I meiocytes. The most enriched term is carbohydrate biosynthesis, indicating that genes related to carbohydrate biosynthesis are preferably targeted by phasiRNAs among all expressed genes in early prophase I meiocytes. The GO term enrichment analysis results in Fig. 3c have been updated.

Throughout, the manuscript is missing information about how replicates were handled for all sequence types and also whether these replicates arise from independent samples (biological replicates rather than technical replicates). It does not appear that variation between replicates was used in the sRNA analysis. For example, the methods state that “read counts of each phasiRNA from wild-type and mutant plants were compared using a Fisher’s exact test. The resulting P value was adjusted to estimate false discovery rate.” A method that uses replicate data (eg, DESeq) is needed to make conclusions regarding change in expression.

Response: All experiments were performed in three biological replicates. We have performed the differential phasiRNA expression analysis by edgeR to further estimate biological variation between three replicates. Fig. 2a and Supplementary Data 4 have been updated.

Similarly, for degradome sequencing, it appears that replicates were merged and treated as a single sample. Identification of sequence tags at the predicted site in multiple replicates would go a long way toward convincing this reviewer that mRNAs

are targeted by phasiRNAs.

Response: Thanks for reviewer's suggestion. Degradome tags identified at predicted site in different biological replicates have been shown in the revised Fig. 3b and Supplementary Data 5.

Finally, for RNAseq data, please be explicit about whether replicate data was entered into Cuffdiff, or whether replicates were merged and assessed as single datasets.

Response: Replicates were not merged as single datasets. Three biological replicates were individually entered into Cuffdiff for differential gene expression. The replicate information was considered when we set the parameters.

Minor comments:

Pg 5, line 23: Just because an AGO contains a conserved catalytic triad (more accurately catalytic tetrad - see Sheng 2014 PBAS), doesn't mean they cleave target mRNAs. Examples include AGO4 from plants; and Twi1p from Tetrahymena. "Slicing" activity could just be a mechanism of AGO loading.

Response: We have revised the sentence to make it accurate. It is required to contain a catalytic tetrad for target mRNA cleavage. which prompt us to test whether phasiRNAs direct target cleavage. Our results indicate that 21-nt phasiRNAs indeed mediate cleavage. By the way, AGO4 uses its slicer activity not only for siRNA loading but also for target RNA cleavage⁸.

It is not clear exactly what is shown in Figure 4c. It appears to be FPKM from the sequencing, but there is no description of the error bars. Given that there must be 3 data points for each bar, it is more appropriate to show these datapoints rather than an average;

Response: In Fig. 4c, the FPKM values showing in each bar plot are from RNA-seq. The error bars represent standard deviations of three biological replicates. The datapoints have been shown in the revised figure.

Although the paper is generally very well written and clear, there are a few typos throughout. Eg, Line 11: "direct cleavage of cleave hundreds of target genes through mRNA cleavage"

Response: We have corrected the grammar mistakes and typos.

References

1. Komiya, R. *et al.* Rice germline-specific Argonaute MEL1 protein binds to phasiRNAs generated from more than 700 lincRNAs. *Plant J* **78**, 385-97 (2014).
2. Fan, Y. *et al.* PMS1T, producing phased small-interfering RNAs, regulates

- photoperiod-sensitive male sterility in rice. *Proc Natl Acad Sci U S A* **113**, 15144-15149 (2016).
3. Wu, H.J., Ma, Y.K., Chen, T., Wang, M. & Wang, X.J. PsRobot: a web-based plant small RNA meta-analysis toolbox. *Nucleic Acids Research* **40**, W22-W28 (2012).
 4. Addo-Quaye, C., Eshoo, T.W., Bartel, D.P. & Axtell, M.J. Endogenous siRNA and miRNA targets identified by sequencing of the Arabidopsis degradome. *Curr Biol* **18**, 758-762 (2008).
 5. Addo-Quaye, C., Miller, W. & Axtell, M.J. CleaveLand: a pipeline for using degradome data to find cleaved small RNA targets. *Bioinformatics* **25**, 130-1 (2009).
 6. Fei, Q., Yang, L., Liang, W., Zhang, D. & Meyers, B.C. Dynamic changes of small RNAs in rice spikelet development reveal specialized reproductive phasiRNA pathways. *J Exp Bot* **67**, 6037-6049 (2016).
 7. Kakrana, A. *et al.* Plant 24-nt reproductive phasiRNAs from intramolecular duplex mRNAs in diverse monocots. *Genome Res* **28**, 1333-1344 (2018).
 8. Qi, Y. *et al.* Distinct catalytic and non-catalytic roles of ARGONAUTE4 in RNA-directed DNA methylation. *Nature* **443**, 1008-12 (2006).

REVIEWER COMMENTS

Reviewer #1 (Remarks to the Author):

Dear authors,

Thanks for the diligent modification of the manuscript and satisfying additional experiments and analysis. The manuscript has improved from its already good shape, and now includes additional important aspects not touched on before. The few comments in the attached pdf files are minor corrections or edits I would suggest. Other than that, I am looking forward to seeing this work published and followed up on in the future.

PS: When I checked the NCBI GEO Accession possibility, it was denied, but I do suspect this was on purpose and will be changed when the manuscript is accepted.

Reviewer #2 (Remarks to the Author):

Compared to the previous version, I believe that the manuscript from Jiang et al has been significantly improved. In particular, the degradome analysis from the *rdr6* mutant has provided a strong support for the authors claims, helping to validate many of the putative phasiRNA targets. I have only a few minor comments/suggestions for the authors consideration:

- I think the authors could include the TAS3 analysis made for *osrdr6-2* (figure 4 in their rebuttal) in Figure S2 and mentioned it in the text, especially in the context of explaining the unexpected high levels of RDR6-independent phasiRNAs in the mutant. Otherwise, readers could interpret this as an indication that some of these phasiRNAs might be generated by an alternative pathway that does not rely on RDR6, which, based on the TAS3-derived tasiRNA levels, doesn't seem to be the case.

- The analysis provided in figure 2 of the rebuttal could also be included in the manuscript. In addition, as a reader, I would be very interested to know if there is any phasiRNA (or PHAS locus) more dominant than others and therefore, having a stronger role in the regulation of several genes. These extra analyses, despite not changing the authors conclusions, could enrich even further the quality of their work.

- In the analysis of the target expression, the authors mentioned that many did not have fold changes above 1.5, but changes were still significant. It would be interesting to mention/know to how many genes this apply.

- Regarding the targets showing high fold changes, are they enriched for transcripts targeted by more than one phasiRNA or by abundantly expressed ones? In the same line, are targets without changes in expression targeted more often by the RDR6-independent pool of phasiRNAs? These could help to explain the low proportion of targets without high changes in expression.

- I would also mention in the manuscript the authors analysis showing that the enriched pathways are not different between the overall targets and the ones showing de-repression in the mutants, since these are the group of transcripts with stronger support for being real targets of the phasiRNAs.

- In the discussion about the possible reasons resulting in low number of targets with de-repress expression (lines 233-242), I would re-write the fourth reason (line 240), including "these" in front of "phasiRNAs" and replacing "should" by "could", since targeting a single transcript could still have significant effect on its expression.

Reviewer #3 (Remarks to the Author):

My primary concerns with the first draft of this paper were that many of the conclusions were confirmatory or poorly-supported by the data presented. Although the paper is improved, in my opinion the authors have not fully addressed these issues.

The early parts of the paper (characterization of phasiRNAs at different developmental stages) does not adequately address the previous work in this area by other laboratories, a point that was also raised by Reviewer 2 and which has not been addressed in the revision. The authors need to be completely clear in their introduction about what is already known about developmental dynamics and genetic analysis of rice phasiRNAs. It is not a problem to begin a paper with confirmatory findings, especially to establish the quality of the libraries for later analysis, but appropriate referencing and attribution are essential.

Clearly the more interesting aspect of the paper is identification of phasiRNA targets in rice. However, when the authors state in their rebuttal letter that "this is the first time to identify phasiRNA targets and to show phasiRNA direct target cleavage", they are incorrect. Phased siRNAs are known to cleave mRNA targets - we call these tasiRNAs since they are known to act in trans. Perhaps any phasiRNA which the authors demonstrate can cleave a target mRNA should be reclassified as a tasiRNA, but the nomenclature is terrible and I'm happy to stick with phasiRNA. Whatever they're called, this paper is not novel in providing examples of phasiRNAs cleaving mRNA targets. Again, the manuscript does not sufficiently discuss the previous research in this area and properly contextualize the results.

Where this paper makes its most significant contribution is to catalog potential targets of the high levels of phasiRNAs found during pollen development. I appreciate the increased stringency in selection of "targeted" mRNAs (which resulted in one-third fewer degradome targets). However, I remain concerned that the novel low-input degradome method has not been benchmarked to determine its sensitivity or specificity. It is not sufficient to say that "it is impossible to isolate large amount of RNA from pure populations of male germ cells. Thus, we are unable to compare our results with those obtained from traditional degradome sequencing." The correct way to benchmark a low-input protocol is to use a tissue from which you *can* collect large amounts of RNA and see whether you get the same results with low input. I will leave it with the editorial team to determine whether this reaches the level of quality desired for a publication at Nature Comms.

A more significant problem is that the revised text still includes this statement, which is unsupported by the data: "The majority of 21-nt phasiRNA-guided cleavage products were detected only in early prophase I meiocytes." Technically, the statement is correct, but only because prophase I meiocytes were sequenced much more thoroughly than other tissues. The revised manuscript now depicts normalized data (reads per 10 million mapped reads) rather than count data, but this does not negate the fact that the underlying data is count based! There are many more degradome reads for prophase I meiocytes than other samples, making it more likely to identify targets in this tissue. According to Figure 3B, the vast majority of these targets have degradome tag abundances of <2 ($=1$ on the \log_2 scale, although is this really a \log_2 scale?). That means fewer than 2 reads per 10 million. But the tetrad libraries have 3.7-9 million reads each, so they have not been sequenced deeply enough to determine whether they cleavage products are present. Additionally, there might be a set of phasiRNA targets that are *specific* to those smaller libraries but they will fall below the threshold for detection. (I also am very confused about the scale in this figure. Why use a \log_2 scale where there is so little range in the data (only up to $\log_2(x)=2.5$)? How can there be so much white on the heat map when log scales cannot plot 0? The legend says a pseudo count was added to each target, but that would create an abundance of >0.5 RP10M in tetrad rep #2.) I realize this is merely one panel in the paper, but it suggests the authors are forcing their data to fit their hypothesis rather than vice versa.

In a similar vein, why only report the targets that are de-repressed in *rdr1* or *mel1* backgrounds? Including data on genes that were downregulated in these backgrounds will help the reader determine whether they believe the title of the manuscript: "21-nt phasiRNAs direct target mRNA cleavage in rice male germ cells". For example, another reading of this result is that of 253 proposed targets of phasiRNAs, 25 go up a little in *mel1*, but there is no evidence for expression for 33 of them. This cherry-picking of results is partly why I requested the comparison with total number of differentially expressed genes in the mutant samples. That comparison should not be relegated to a response to reviewers, but included in the manuscript for readers to interpret. Perhaps consider a volcano plot of differential expression, with the predicted phasiRNA targets in a different color?

I'd like to end on a positive note. This paper has a great set of siRNA sequencing and a promising start on a technically challenging experiment (low-input degradome sequencing). I think the data also support the conclusion that although there are loads of phasiRNAs made during pollen development, there is no strong evidence that these are important regulators of gene expression. A paper that states that might not be as flashy, but it would be a stronger contribution to the literature.

We appreciate the constructive comments made by the reviewers. We have provided additional data and revised our manuscript to address the concerns raised by the reviewers. We wish the revisions are sufficient and the manuscript is now acceptable for publication. Point-by-point responses are listed below.

Reviewer #1:

Dear authors,

Thanks for the diligent modification of the manuscript and satisfying additional experiments and analysis. The manuscript has improved from its already good shape, and now includes additional important aspects not touched on before. The few comments in the attached pdf files are minor corrections or edits I would suggest. Other than that, I am looking forward to seeing this work published and followed up on in the future.

PS: When I checked the NCBI GEO Accession possibility, it was denied, but I do suspect this was on purpose and will be changed when the manuscript is accepted.

Response: We thank the reviewer for the positive comments. We have revised our manuscript as suggested. We have uploaded our data to NCBI GEO (accession number GSE149800). The data will be released to the public as soon as our manuscript is published. To get access to our data while the manuscript is under review, please use the link <https://www.ncbi.nlm.nih.gov/geo/query/acc.cgi?acc=GSE149800> and the token epeloqkwtrizvqh.

Reviewer #2:

Compared to the previous version, I believe that the manuscript from Jiang et al has been significantly improved. In particular, the degradome analysis from the rdr6 mutant has provided a strong support for the authors claims, helping to validate many of the putative phasiRNA targets. I have only a few minor comments/suggestions for the authors consideration:

- I think the authors could include the TAS3 analysis made for osrdr6-2 (figure 4 in their rebuttal) in Figure S2 and mentioned it in the text, especially in the context of explaining

the unexpected high levels of RDR6-independent phasiRNAs in the mutant. Otherwise, readers could interpret this as an indication that some of these phasiRNAs might be generated by an alternative pathway that does not rely on RDR6, which, based on the TAS3-derived tasiRNA levels, doesn't seem to be the case.

Response: We thank the reviewer for the suggestion. We have included the *TAS3* analysis in Supplementary Fig. 2 and mentioned it in the main text.

- The analysis provided in figure 2 of the rebuttal could also be included in the manuscript. In addition, as a reader, I would be very interested to know if there is any phasiRNA (or PHAS locus) more dominant than others and therefore, having a stronger role in the regulation of several genes. These extra analyses, despite not changing the authors conclusions, could enrich even further the quality of their work.

Response: We thank the reviewer for the suggestion. We have included figure 2 of the rebuttal in Supplementary Fig. 9. As shown in Supplementary Fig. 9, the majority of *PHAS* loci (88.8%) have only one target. Only a few *PHAS* loci have more than one target. One example is *21PHAS503_184*. It can target 23 genes including 3 expressed genes and 20 transposons, which belong to one transposon family (Supplementary Data 5). Future investigations are required to validate this result.

- In the analysis of the target expression, the authors mentioned that many did not have fold changes above 1.5, but changes were still significant. It would be interesting to mention/know to how many genes this apply.

Response: There are 9 and 23 genes, which showed significant changes but did not have fold changes above 1.5 in *osrdr6-2* and *mell-4*, respectively. We have included this in the revised manuscript.

- Regarding the targets showing high fold changes, are they enriched for transcripts targeted by more than one phasiRNA or by abundantly expressed ones? In the same line, are targets without changes in expression targeted more often by the RDR6-independent pool of phasiRNAs? These could help to explain the low proportion of targets without

high changes in expression.

Response: Among 65 significantly derepressed targets in early prophase I meicytes of *rdr6-2* and/or *mel1-4*, only 6 targets are targeted by ≥ 2 phasiRNAs. But they did not show higher fold change than other phasiRNA targets (Supplementary Data 7). Therefore, targets showing high fold changes are not enriched for transcripts targeted by more than one phasiRNAs. Our analysis revealed that the correlation between fold change of target expression and phasiRNA abundance is weak ($Cor=0.02$) (Fig. 1a below). Therefore, targets showing high fold changes are not enriched for transcripts targeted by abundant phasiRNAs. Most targets without changes in expression are targeted by *OSRDR6*-dependent phasiRNAs (Fig. 1b below). They do not undergo changes in expression possibly because there are feedback regulatory mechanisms and phasiRNAs may regulate target gene expression using other modes of action. Furthermore, phasiRNAs may only induce subtle changes of target gene expression. However, the subtle changes could not pass the fold-change cut-off.

Figure 1. Relationship between fold change of target expression and abundance and fold change of phasiRNA. **a**, Scatter plot showing target expression fold change and phasiRNA abundance. **b**, Scatter plot showing target expression fold change and phasiRNA abundance fold change. Red dots represent significantly upregulated phasiRNA target genes, grey dots represent unchanged phasiRNA target genes.

- I would also mention in the manuscript the authors analysis showing that the enriched pathways are not different between the overall targets and the ones showing de-repression in the mutants, since these are the group of transcripts with stronger support for being real targets of the phasiRNAs.

Response: We thank the reviewer for the suggestion. We performed GO analysis on derepressed targets. The results showed that the derepressed targets are also enriched for the carbohydrate biosynthetic pathway (Supplementary Fig. 13). We have included this in the revised manuscript.

In the discussion about the possible reasons resulting in low number of targets with de-repress expression (lines 233-242), I would re-write the fourth reason (line 240), including “these” in front of “phasiRNAs” and replacing “should” by “could”, since targeting a single transcript could still have significant effect on its expression.

Response: We have re-written the sentence in the revised manuscript.

Reviewer #3:

My primary concerns with the first draft of this paper were that many of the conclusions were confirmatory or poorly-supported by the data presented. Although the paper is improved, in my opinion the authors have not fully addressed these issues.

The early parts of the paper (characterization of phasiRNAs at different developmental stages) does not adequately address the previous work in this area by other laboratories, a point that was also raised by Reviewer 2 and which has not been addressed in the revision. The authors need to be completely clear in their introduction about what is already known about developmental dynamics and genetic analysis of rice phasiRNAs. It is not a problem to begin a paper with confirmatory findings, especially to establish the quality of the libraries for later analysis, but appropriate referencing and attribution are essential.

Response: We thank the reviewer for the suggestion. We have introduced developmental

dynamics and genetic analysis of rice phasiRNAs in the revised manuscript. Previous findings from other laboratories have been summarized and cited.

Clearly the more interesting aspect of the paper is identification of phasiRNA targets in rice. However, when the authors state in their rebuttal letter that “this is the first time to identify phasiRNA targets and to show phasiRNA direct target cleavage”, they are incorrect. Phased siRNAs are known to cleave mRNA targets - we call these tasiRNAs since they are known to act in trans. Perhaps any phasiRNA which the authors demonstrate can cleave a target mRNA should be reclassified as a tasiRNA, but the nomenclature is terrible and I’m happy to stick with phasiRNA. Whatever they’re called, this paper is not novel in providing examples of phasiRNAs cleaving mRNA targets. Again, the manuscript does not sufficiently discuss the previous research in this area and properly contextualize the results.

Response: We thank the reviewer for the comment. What we meant is that targets and action mode of reproductive phasiRNAs in grasses are revealed for the first time. We agree with this reviewer that phasiRNAs could be classified as ta-siRNAs if they act in trans. To avoid further confusion about the nomenclature of ta-siRNA and phasiRNA, we would like to stick with phasiRNA even we have shown that rice reproductive phasiRNAs do act in trans. Although reproductive phasiRNAs have been suggested to regulate male fertility, previous studies failed to find their targets and their mode of action was poorly understood. We believe that our findings provide important insights into the role of these phasiRNAs in plant reproduction.

Where this paper makes its most significant contribution is to catalog potential targets of the high levels of phasiRNAs found during pollen development. I appreciate the increased stringency in selection of “targeted” mRNAs (which resulted in one-third fewer degradome targets). However, I remain concerned that the novel low-input degradome method has not been benchmarked to determine it’s sensitivity or specificity. It is not sufficient to say that “it is impossible to isolate large amount of RNA from pure

populations of male germ cells. Thus, we are unable to compare our results with those obtained from traditional degradome sequencing.” The correct way to benchmark a low-input protocol is to use a tissue from which you *can* collect large amounts of RNA and see whether you get the same results with low input. I will leave it with the editorial team to determine whether this reaches the level of quality desired for a publication at Nature Comms.

Response: We thank the reviewer for the suggestion. As suggested, we applied the low-input protocol (with 5 ng and 50 ng of total RNA) and the regular PARE protocol (with 150 µg total RNA) to 4-week-old rice seedlings. We identified miRNA targets using the same criteria we used for phasiRNA target identification. 131, 128 and 159 miRNA targets were identified by our low-input protocol with 5 ng and 50 ng of total RNA and the regular PARE method with 150 µg of total RNA, respectively (Supplementary Fig. 5). Among these, 130, 125 and 158 are previously validated miRNA targets in rice seedlings (Li et al., 2010; Wu et al., 2009; Zhou et al., 2010). The miRNA targets identified by our low-input protocol and the PARE protocol overlapped to a great extent. 17 and 19 miRNA targets were identified by our low-input protocol with 5 ng and 50 ng of total RNA, but not identified by the regular PARE protocol. 14 and 18 among the 17 and 19 miRNA targets were previously validated miRNA targets. These data suggest that our low-input protocol has a specificity and sensitivity comparable to the regular PARE protocol.

A more significant problem is that the revised text still includes this statement, which is unsupported by the data: “The majority of 21-nt phasiRNA-guided cleavage products were detected only in early prophase I meiocytes.” Technically, the statement is correct, but only because prophase I meiocytes were sequenced much more thoroughly than other tissues. The revised manuscript now depicts normalized data (reads per 10 million mapped reads) rather than count data, but this does not negate the fact that the underlying data is count based! There are many more degradome reads for prophase I meiocytes than

other samples, making it more likely to identify targets in this tissue. According to Figure 3B, the vast majority of these targets have degradome tag abundances of <2 ($=1$ on the \log_2 scale, although is this really a \log_2 scale?). That means fewer than 2 reads per 10 million. But the tetrad libraries have 3.7-9 million reads each, so they have not been sequenced deeply enough to determine whether they cleavage products are present. Additionally, there might be a set of phasiRNA targets that are **specific** to those smaller libraries but they will fall below the threshold for detection. (I also am very confused about the scale in this figure. Why use a \log_2 scale where there is so little range in the data (only up to $\log_2(x)=2.5$)? How can there be so much white on the heat map when log scales cannot plot 0? The legend says a pseudo count was added to each target, but that would create an abundance of >0.5 RP10M in tetrad rep #2.) I realize this is merely one panel in the paper, but it suggests the authors are forcing their data to fit their hypothesis rather than vice versa.

Response: We thank the reviewer for the comment. To address the problem, the tetrad and microspore libraries were sequenced at depth higher than the early prophase I meiocyte libraries (Supplementary Data 1). Data analysis revealed that the majority of 21-nt phasiRNA-guided cleavage products were still only detected in early prophase I meiocytes (Fig. 3b). This result is consistent with the high-level accumulation of 21-nt phasiRNAs and MEL1 in early prophase I meiocytes.

Sorry about the confusion about Fig. 3b. The counts of degradome tags at +1 positions of 21-nt phasiRNA-guided cleavage sites in early prophase I meiocytes, tetrads and microspores varied in a wide range (from 0 to 1500). The counts of degradome tags at +1 positions of 21-nt phasiRNA-guided cleavage sites in early prophase I meiocytes were high. Their $\log_2(\text{RP10M}+1)$ values were often ≥ 5 . In contrast, the counts of degradome tags at +1 positions of 21-nt phasiRNA-guided cleavage sites in tetrads and microspores were low. Their $\log_2(\text{RP10M}+1)$ values were often ≤ 2 (detailed in Supplemental Data 5). To visualize the abundances of degradome tags in tetrads and

microspores better, we previously chose a narrow color intensity scale 0-2.5 in Fig. 3b. The white color (color intensity = 0) indicates that the RP10M values of indicated targets are equal to 0 ($\log_2(0+1) = 0$). When the value of $\log_2(\text{RP10M}+1)$ is ≥ 2.5 , we used the same intensity of red color. In the revised Fig. 3b, the color scale has been changed to 0-10.

In a similar vein, why only report the targets that are de-repressed in *rdr1* or *mell1* backgrounds? Including data on genes that were downregulated in these backgrounds will help the reader determine whether they believe the title of the manuscript: “21-nt phasiRNAs direct target mRNA cleavage in rice male germ cells”. For example, another reading of this result is that of 253 proposed targets of phasiRNAs, 25 go up a little in *mell1*, but there is no evidence for expression for 33 of them. This cherry-picking of results is partly why I requested the comparison with total number of differentially expressed genes in the mutant samples. That comparison should not be relegated to a response to reviewers, but included in the manuscript for readers to interpret. Perhaps consider a volcano plot of differential expression, with the predicted phasiRNA targets in a different color?

Response: We thank the reviewer for the comment. We have reported the numbers of targets that are downregulated. We have made volcano plots showing differential expression of phasiRNA targets and other expressed genes in *osrdr6-2* and *mell1-4* (Supplementary Fig. 12).

I'd like to end on a positive note. This paper has a great set of siRNA sequencing and a promising start on a technically challenging experiment (low-input degradome sequencing). I think the data also support the conclusion that although there are loads of phasiRNAs made during pollen development, there is no strong evidence that these are important regulators of gene expression. A paper that states that might not be as flashy, but it would be a stronger contribution to the literature.

Response: We thank the reviewer for the positive comment. We think that our degradome

data have shown that hundreds of genes are targeted by 21-nt phasiRNAs, which will serve a new starting point for further investigation on the functions of phasiRNAs and their targets in regulating male fertility in rice.

References:

Li, Y.F., Zheng, Y., Addo-Quaye, C., Zhang, L., Saini, A., Jagadeeswaran, G., Axtell, M.J., Zhang, W., and Sunkar, R. (2010). Transcriptome-wide identification of microRNA targets in rice. *Plant J* 62, 742-759.

Wu, L., Zhang, Q., Zhou, H., Ni, F., Wu, X., and Qi, Y. (2009). Rice MicroRNA effector complexes and targets. *Plant Cell* 21, 3421-3435.

Zhou, M., Gu, L., Li, P., Song, X., Wei, L., Chen, Z., and Cao, X. (2010). Degradome sequencing reveals endogenous small RNA targets in rice (*Oryza sativa* L. ssp. indica). *Frontiers in Biology* 5, 67-90.

REVIEWERS' COMMENTS

Reviewer #2 (Remarks to the Author):

I believe that the authors have properly addressed all my concerns/comments and therefore, I am happy to recommend this manuscript for publication at Nature communications.

Reviewer #3 (Remarks to the Author):

Although I continue to disagree with some of the interpretations in this manuscript, the authors have now including sufficient analysis to allow readers to judge for themselves.